# A rice Serine/Threonine receptor-like kinase regulates arbuscular mycorrhizal symbiosis at the peri-arbuscular membrane

Ronelle Roth [1], Marco Chiapello [2,3], Héctor Montero [1], Peter Gehrig[4], Jonas Grossmann[4], Kevin O'Holleran[5], Denise Hartken[1], Fergus Walters[1], Shu-Yi Yang[1], Stefan Hillmer[6], Karin Schumacher[7], Sarah Bowden [8], Melanie Craze[8], Emma J. Wallington [8], Akio Miyao [9], Ruairidh Sawers[10], Enrico Martinoia[11] & Uta Paszkowski [1,2]

In terrestrial ecosystems most plant species live in mutualistic symbioses with nutrient-delivering arbuscular mycorrhizal (AM) fungi. Establishment of AM symbioses includes transient, intracellular formation of fungal feeding structures, the arbuscules. A plant-derived peri-arbuscular membrane (PAM) surrounds the arbuscules, mediating reciprocal nutrient exchange. Signaling at the PAM must be well coordinated to achieve this dynamic cellular intimacy. Here, we identify the PAM-specific Arbuscular Receptor-like Kinase 1 (ARK1) from maize and rice to condition sustained AM symbiosis. Mutation of rice *ARK1* causes a significant reduction in vesicles, the fungal storage structures, and a concomitant reduction in overall root colonization by the AM fungus *Rhizophagus irregularis*. Arbuscules, although less frequent in the *ark1* mutant, are morphologically normal. Co-cultivation with wild-type plants restores vesicle and spore formation, suggesting ARK1 function is required for the completion of the fungal life-cycle, thereby defining a functional stage, post arbuscule development.

[1] Department of Plant Sciences, University of Cambridge, Cambridge CB2 3EA, UK. [2] Department of Plant Molecular Biology, University of Lausanne, Biophore, 1015 Lausanne, Switzerland. [3] Department of Biochemistry, University of Cambridge, Cambridge CB2 1QW, UK. [4] Functional Genomics Center, University and ETH Zürich, Winterthurerstr. 190, 8057 Zürich, Switzerland. [5] Cambridge Advanced Imaging Centre, University of Cambridge, Cambridge CB2 3DY, UK. [6] Electron Microscopy Core Facility, University of Heidelberg, Im Neuenheimer Feld 345, 69120 Heidelberg, Germany. [7] Centre for Organismal Studies, University of Heidelberg, Im Neuenheimer Feld 230, 69120 Heidelberg, Germany. [8] The John Bingham Laboratory, National Institute of Agricultural Botany, Huntingdon Road, Cambridge CB3 0LE, UK. [9] National Agriculture and Food Research Organization, Advanced Genomics Breeding Section, Institute of Crop Science, 2–1–2, Kannondai, Tsukuba, Ibaraki 305-8518, Japan. [10] Laboratorio Nacional de Genómica para la Biodiversidad, Centro de Investigación y de Estudios Avanzados, 36821 Irapuato, GTO, Mexico. [11] Institute of Plant Biology, University of Zürich, Zollikerstrasse 107, 8008 Zürich, Switzerland. These authors contributed equally: Ronelle Roth, Marco Chiapello. Correspondence and requests for materials should be addressed to R.R. (email: rr472@cam.ac.uk) or to U.P. (email: up220@cam.ac.uk)

Arbuscular mycorrhizal (AM) symbioses are mutually beneficial interactions between soil fungi of the Glomeromycotina and most land plants including the important cereals maize and rice. AM fungi provide plants with water and essential minerals such as inorganic phosphate (Pi) and nitrogen, and are thereby of central importance for (agro) ecosystem productivity. AM fungi are fatty acid heterotrophs[1] that depend on host-delivered organic carbon (C) in the form of fatty acids[2–5] in order to complete their life cycle.

The primary sites of symbiotic nutrient exchange are highly branched fungal feeding structures, called arbuscules that form within the cells of the root inner cortex. Arbuscules are short-lived, developing and collapsing within days. Arbuscule formation is accompanied by massive de novo synthesis of the plant peri-arbuscular membrane (PAM) that envelops the fungal structure throughout its life span. Fungus-provided Pi and likely also plant-delivered fatty acids are exchanged across the PAM via phosphate uptake transporters of the PT11/PT4 class[6–10] and the putative lipid exporters, Stunted Arbuscules 1 and 2 (STR1/STR2) of the ABCG family of transporters, respectively[11,12]. At the whole root level, arbuscule development occurs non-synchronously, with the simultaneous presence of all different stages from initiation to collapse. Conversely, storage bodies such as vesicles and spores increase over time as the fungus assimilates C and reaches the completion of the fungal life cycle[13,14].

The intracellular development of arbuscules reflects a fine-tuned coordination of cellular activities of both symbiotic partners. Due to the genetic intractability of the fungus, little is known about the molecular changes that occur during the differentiation of simple hyphal filaments into highly branched arbuscules. In plants, recent studies have revealed that complex cell-autonomous transcriptional programs are activated in the root cortex to accommodate fungal arbuscules. Furthermore, distinct transcriptional signatures are associated with specific stages of arbuscule development and collapse, indicating that plant gene expression is tightly coordinated with fungal development (reviewed in ref. [15] and references therein). Such coordination requires the action of complex signaling pathways to control transcriptional modules. The common symbiosis signaling pathway, which is additionally required for nodulation symbiosis with nitrogen-fixing bacteria[16,17] plays a key role in plant transcriptional regulation throughout arbuscule development[18–23].

To identify putative PAM-specific signaling proteins from *Rhizophagus irregularis* colonized roots, we performed an innovative membrane proteome analysis combined with characterization of the transcriptome of laser-dissected arbusculated cells. Our analysis identified two putative signaling proteins that were specifically induced in arbuscule-containing cells, a Lys-M domain receptor-like kinase (LYK1/RLK3) and a serine/threonine receptor-like kinase (Ser/Thr RLK), named ARK1 for Arbuscular Receptor-like Kinase 1. ARK1 was shown to be localized specifically to the PAM. Moreover, we present evidence that ARK1 is required for the formation of fungal storage vesicles by playing an essential role in sustaining fungal growth through to completion of its life cycle. The formation of fully developed arbuscules does not require ARK1, indicating that ARK1 functions post-arbuscule development, playing a critical role in the maintenance of AM symbiosis. Together, our results suggest a role for ARK1 in signaling at the PAM and propose ARK1 function to be essential for the support of fungal fitness.

## Results

### The membrane proteome of mycorrhizal maize and rice roots.
The combination of the low abundance of PAM relative to other membranes and the nonsynchronous formation of arbuscules in the root (Supplementary Figure 1a, c) has greatly hampered approaches to comprehensively identify PAM proteins in any plant species. Here, we exploited the accumulation of yellow apocarotenoid[24,25] pigment associated with the presence of arbuscules in the roots of maize to collect samples enriched in arbuscules, and thus PAM-containing tissue for proteomic analysis.

We designed our own proteomics workflow (Supplementary Figure 2, for details see Methods) and estimated the sensitivity of the workflow by selecting a number of marker proteins that occur in specific cell-types and/or subdomains, and can therefore be considered low abundant (Supplementary Fig. 1a−c). The iron-phytosiderophore transporter YS1 localizes to the plasma-membrane (PM) of rhizodermal cells (Supplementary Fig. 1a)[26], whereas the Mn and Cd transporter NRAMP5 polarly resides within the distal membrane (EDPM, Supplementary Fig. 1b) and CASP1 within the Casparian Strip subdomain (CSD, Supplementary Fig. 1b) of endo- and exodermal cells, respectively[27,28]. Furthermore, maize PT6[29] belongs to the group of conserved symbiotic phosphate transporters that reside within the PAM surrounding arbuscular fine branches but are absent from other membranes (Supplementary Fig. 1c)[7,9] and therefore serves as an excellent marker for the PAM subdomain[30]. Similarly, the ABCG-transporters STR1 and STR2 reside within fine-branch enveloping PAM subdomains (Supplementary Fig. 1c)[11], but have previously not been detected in proteomics approaches[30].

The maize rhizodermal plasma membrane marker YS1 (GRMZM2G156599) was identified by ten and eight unique peptides in mock inoculated plasma membrane (MPM) and Arbuscular Plasma Membrane (APM) proteome (Table 1); likewise, the marker for exo-/endodermal distal membrane domain, NRAMP5 (GRMZM2G147560) was represented by two and four peptides, respectively (Table 1). Remarkably, CASP1-like protein (GRMZM2G382104) was consistently detected by two unique peptides across all mock and mycorrhizal replicates (Table 1). Gene ontology numbers were used to detect proteins from different compartments using the Blast2Go software[31], which further confirmed the PM enrichment relative to other membrane proteins. The total of 63.6% of the identified membrane proteins for each mock and mycorrhizal roots were assigned to the PM[32,33] whereas the remaining 36.4% were predicted to localize to the endoplasmic reticulum (ER)[34,35], the tonoplast (VAC)[36,37] or the mitochondrial (MIT)[38,39] membrane (Supplementary Fig. 2a, Supplementary Table 1). Together, these proof-of-concept efforts confirmed the suitability of this workflow for the identification of underrepresented membrane proteins from maize root tissue (Supplementary Data 1).

A total of 3640 membrane or membrane-associated proteins (hereafter named membrane proteins for simplicity) were obtained from mock inoculated (MI) and 4192 membrane proteins including 624 fungal proteins isolated from *R. irregularis* colonized maize roots (MYC, Supplementary Table 2, Supplementary Data 2). Comparison of MYC or MI roots identified 722 and 776 maize proteins that were specific to mycorrhizal (ZmAPM) or control roots (ZmMPM), respectively (Fig. 1a, Supplementary Table 2, Supplementary Data 2). In addition, a total of 2846 membrane proteins were detected in both MYC and MI roots, of which 177 were detected to a lower (>2-fold) level, and 83 were detected to a higher (>2-fold) level in MYC compared with MI roots (Fig. 1a, Supplementary Table 2, Supplementary Data 2). The PAM marker ZmPT6 (GRMZM5G881088) was detected by 18 unique peptides (Table 1). Similarly, the maize homologs of the PAM-localized ABC transporters ZmSTR1 (GRMZM2G357034) and ZmSTR2 (GRMZM2G035276)[11,12] were reliably identified by 15 and 18

**Table 1 Identification of cell-type-specific marker proteins in mock control and arbuscule-containing maize roots**

| Accession number | Description | Membrane subdomain | Highest number of unique peptides MPM | Quantitative value MPM | Highest number of unique peptides APM | Quantitative value APM | Reference |
|---|---|---|---|---|---|---|---|
| GRMZM2G156599 | Iron-phytosiderophore transporter yellow stripe 1 (YS1) | RPM | 10 | 23.01 | 8 | 22.09 | 26 |
| GRMZM2G147560 | NRAMP-type metal cation transporter 5 (NRAMP5) | EDPM | 2 | 31.49 | 4 | 54.69 | 28 |
| GRMZM2G382104 | CASP1-like protein | CSD | 2 | 53.53 | 2 | 47.40 | 27 |
| GRMZM5G881088 | Inorganic phosphate transporter PT6 | PAM | 0 | 0 | 18 | 88.36 | 29 |
| GRMZM2G357034 | ABC-2 type transporter domain containing protein STR1 | PAM | 0 | 0 | 15 | 16.41 | 11 |
| GRMZM2G035276 | ABC-2 type transporter domain containing protein STR2 | PAM | 0 | 0 | 18 | 26.43 | 11 |

*RPM* rhizodermal plasma membrane, *EDPM* exo- and endodermal plasma membrane, *CSD* Casparian strip membrane domain, *PAM* peri-arbuscular membrane, *MPM* Mock Plasma Membrane Proteome, *APM* Arbuscular Plasma Membrane Proteome

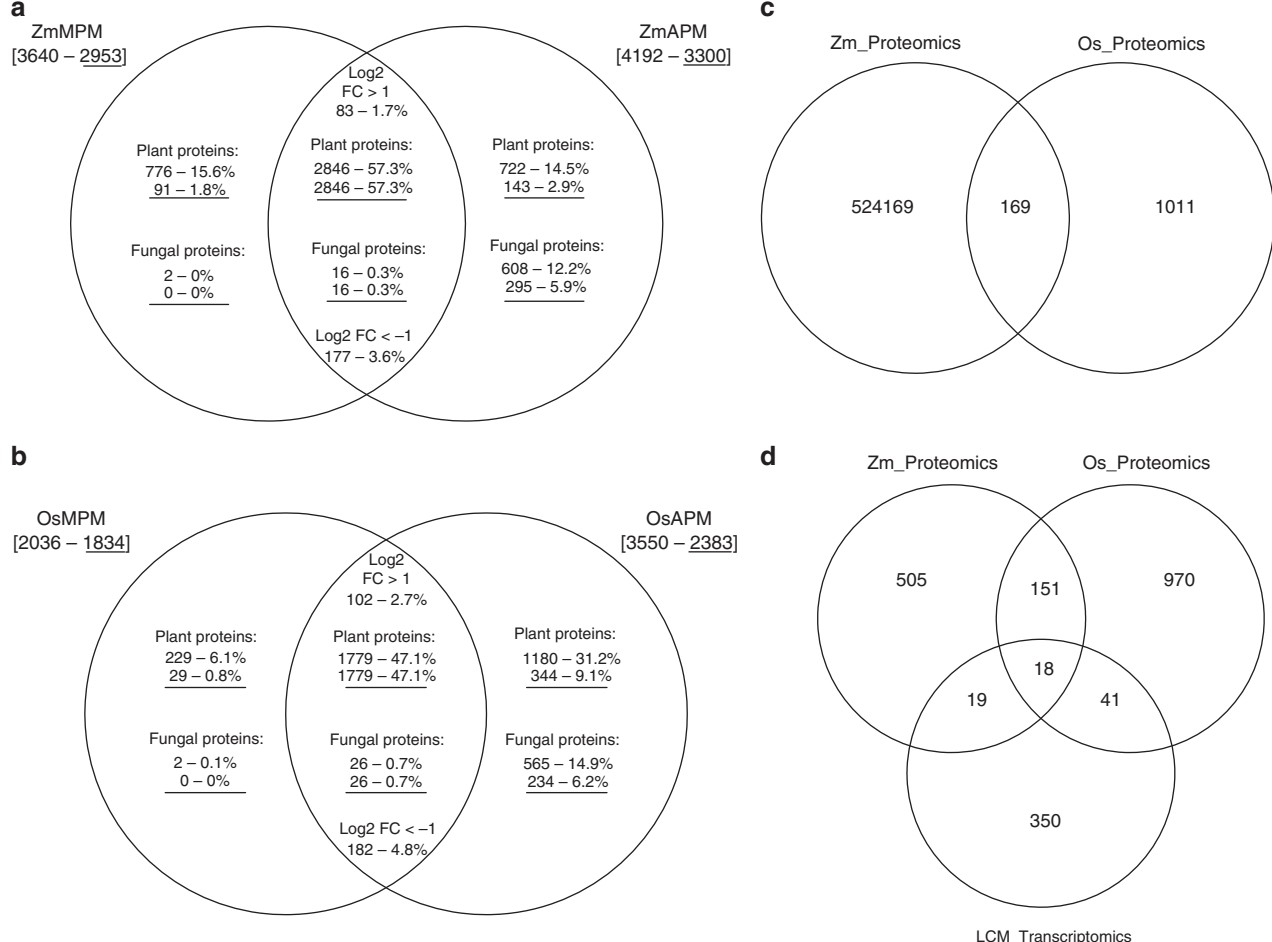

**Fig. 1** Venn diagrams of maize and rice proteome and LCM transcriptome analyses. **a** Number of identified proteins detected in maize mock control (ZmMPM), arbuscule-containing (ZmAPM) or in both root samples. **b** Number of identified proteins detected in rice mock control (OsMPM), arbuscule-containing (OsAPM) or in both root samples. **c** Number of AM symbiosis-associated membrane proteins identified from maize (ZmAPM) or in rice (OsAPM) or in both species. **d** Number of AM symbiosis-associated proteins from maize (ZmAPM) or rice (OsAPM) or arbuscule-associated LCM transcripts (LCM_transcriptomics). Numbers in brackets indicate the total number of detected proteins; underlined numbers indicate the quantified proteins. The table shows the numbers of all plant or fungal proteins

peptides each (Table 1), confirming the significantly improved sensitivity of our proteomics strategy[30,40,41].

Having established a robust protocol for membrane proteomics in maize, we next applied the same procedure to rice roots colonized by *R. irregularis*. Unlike maize, in which all root types are colonized equally, in rice, large lateral roots are preferentially colonized[42], further diluting the presence of colonized sectors across the whole root system. We therefore expected a lower recovery of PAM-bound proteins. To ensure roots well-colonized by *R. irregularis*, total root length colonization had to have reached >70%.

Similar to maize we successfully achieved an enrichment of PM proteins over other endomembrane proteins by continuous sucrose density centrifugation (Supplementary Fig. 2b, Supplementary Table 3, Supplementary Data 1). Confirmation that the rice MPM and APM dataset contained sufficient representation of low abundant cell-type specific markers was obtained by finding eight and six unique peptides from OsYS1 and two and three from OsCASP1-like protein (Supplementary Table 4). For the symbiotic PAM markers ten and seven unique peptides corresponded to OsPT11 (LOC_Os01g46860) and OsSTR1 (LOC_Os09g23640) (Supplementary Table 4). A total of 2036 membrane proteins were obtained from MI and 3550 membrane proteins including 591 fungal proteins were identified from MYC rice roots (Supplementary Tables 5, Supplementary Data 3). Of these, 1180 proteins accumulated exclusively in colonized roots (OsAPM), while 229 membrane proteins were present only in MI roots (OsMPM) (Fig. 1b, Supplementary Data 3). A lower number of symbiosis-associated rice (2959) as opposed to maize (3568) proteins were obtained from the corresponding APM datasets (Supplementary Tables 2 and 5), in line with the pigmentation-based PAM-enrichment effect of the maize samples. Consistently, the maize APM dataset had a higher correlation efficiency ($R = 0.9$) compared to that of rice ($R = 0.75$) (Supplementary Tables 2 and 5). Intersection of the rice and maize APM datasets identified the presence of 169 pairs of orthologous proteins present in the two datasets (Fig. 1c, Supplementary Data 4). The relatively large number of proteins detected in only one species (Fig. 1c) could represent species-specific proteins in addition to technical noise arising from difficulties finding the correct protein match between two species due to e.g. the existence of multiple isoforms.

In summary, presence of PAM-specific proteins in both final maize and rice APM datasets confirmed that both datasets were enriched for low abundant membrane-bound proteins.

**Membrane proteins from arbuscule-containing tissue.** To determine if transcripts encoding the proteins in the shared APM dataset (Supplementary Data 4) were enriched in arbusculated cells, we applied laser capture microdissection (LCM) to rice cortex cells that either contained arbuscules (A), were non-colonized but adjacent to arbusculated cells to monitor systemic induction (S) or were isolated from MI roots (M), and analyzed the resulting transcriptomes by microarray hybridization. Using a cut-off of <1% false discovery rate (FDR) and >2-fold change, a total of 419 transcripts were identified that accumulated differentially in A compared to M or S samples (Supplementary Fig. 3a and 3b). These included 370 transcripts that were upregulated (Supplementary Fig. 3a) and 49 transcripts that were downregulated (Supplementary Fig. 3b, Supplementary Data 5). A considerable overlap was observed between the differentially expressed genes identified and our published analysis of whole roots[43], indicating the robustness of the LCM transcriptome dataset (Supplementary Fig. 3c, Supplementary Data 5).

Next, we separately overlaid the LCM dataset with the maize and rice APM datasets (Fig. 1d, Supplementary Data 6). Of the 722 maize and 1180 rice proteins, 37 and 59, respectively, corresponded to transcripts that accumulated only in arbuscule-containing cells with 18 proteins shared between maize and rice, and included the PAM-marker protein PT11 (Fig. 1d, Table 2, Supplementary Data 6). Interestingly, our final list of 18 arbuscule-associated proteins also included the H⁺ATPase OsHA1, which in *Medicago truncatula* is known to localize to the PAM where it drives the electrochemical proton-gradient needed for mycorrhiza-induced Pi uptake[44,45]. In addition, the ER-localized glycerol-3-phosphate acyltransferase (GPAT) Reduced Arbuscular Mycorrhiza 2 (RAM2) that is required for arbuscule development also appeared in our final dataset[5,46]. As only low levels of organellar contamination had been detected in our PM fractions (Supplementary Fig. 2a, b, Supplementary Table 1 and 3), we attributed the recovery of RAM2 to its high expression level and the proximity of the ER to arbuscular branches. Importantly, among the final list of proteins associated with arbusculated cells were two putative RLKs, the LysM RLK called OsLYK1/OsRLK3 in rice (LOC_Os01g36550.1, called LYK1 throughout the manuscript) and the Ser/Thr RLK ARK1 (LOC_Os11g26140.1), which had earlier been described as the late symbiosis marker, AM14, in mycorrhizal rice[47]. Our approach thus led to the identification of two candidate proteins with potential roles in signaling at the PAM.

**Promoter activity in arbuscule-containing cells.** To examine the spatial expression of *LYK1* and *AM14*, we generated stable transgenic rice *pLYK1:GUS* and *pAM14:GUS* reporter lines. The comparison of GUS-stained roots in *R. irregularis*-inoculated lines confirmed *LYK1* and *AM14* promoters to be active exclusively in arbuscule-containing cells (Fig. 2, Supplementary Fig. 4). According to the specific expression of both RLKs in arbuscule-containing cells, we focused our further investigations on both genes and renamed AM14 Arbuscular Receptor Kinase 1 (ARK1).

**Gene and protein structure of LYK1 and ARK1.** The *LYK1* gene consists of ten exons and encodes a protein of 618 amino acids (aa) (Supplementary Fig. 5a). Based on InterPro predictions LYK1 contains an N-terminal signal peptide (SP) for export, a

### Table 2 Rice proteins present across all datasets

| Gene identifier | Protein description |
| --- | --- |
| LOC_Os12g42230 | Pyruvate dehydrogenase E1 component subunit beta-3 |
| LOC_Os11g26140 | ARK1 \| Arbuscule-specific Receptor-like Kinase 1 |
| LOC_Os04g55060 | KAS III \| 3-ketoacyl-acyl carrier protein |
| LOC_Os01g55940 | OsGH3.2 \| Probable indole-3-acetic acid-amido synthetase |
| LOC_Os01g46860 | PHT1;11 \| phosphate transporter 11 |
| LOC_Os03g52570 | GPAT6 \| glycerol-3-phosphate acyltransferase 6 |
| LOC_Os05g47500 | MDR-like ABC transporter |
| LOC_Os08g05710 | ABC transporter, ATP-binding protein |
| LOC_Os03g01120 | OsHA1 \| H(+)-ATPase 1 |
| LOC_Os06g47130 | C2 domain containing protein |
| LOC_Os01g72710 | Protein of unknown function (DUF3411) |
| LOC_Os01g52750 | OsSub3 \| Putative Subtilisin homolog |
| LOC_Os01g01710 | 1-deoxy-D-xylulose 5-phosphate reductoisomerase |
| LOC_Os09g39530 | Cupin-domain containing protein |
| LOC_Os08g42590 | MtN19 \| Nod19 stress upregulated protein |
| LOC_Os01g46390 | DUF538 domain containing protein |
| LOC_Os05g51240 | Hydrolase, alpha/beta fold family domain containing protein |
| LOC_Os01g36550 | LYK1 \| LysM-domain containing protein kinase 1 |

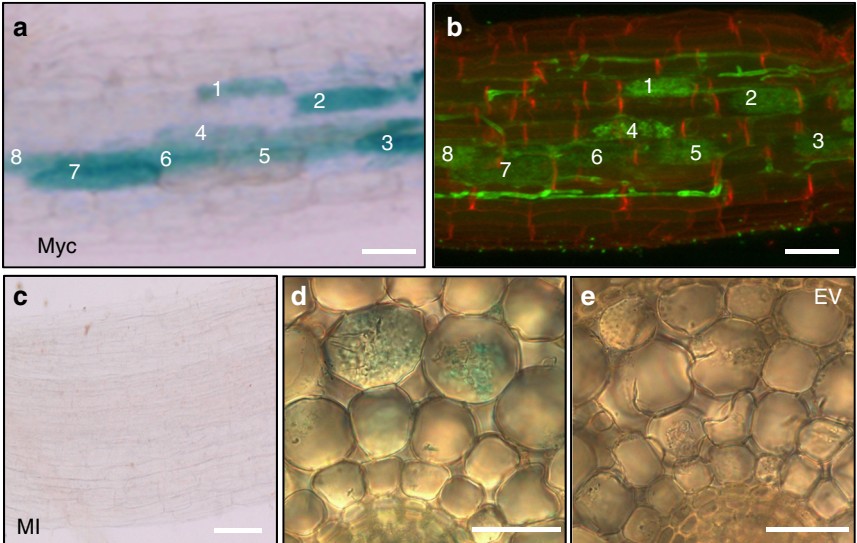

**Fig. 2** Promoter activity of *ARK1* in *R. irregularis* colonized roots. **a** GUS staining in cortical cells of *R. irregularis* colonized (Myc) *pARK1:GUS* roots containing arbuscules (numbered 1–8) but not in nonarbusculated cells at 6 weeks post-inoculation. **b** Wheat germ agglutinin (WGA)-Alexa Fluor 488-stained root sector from (**a**) showing intercellular hyphae and arbuscules (numbered 1–8). **c** Mock-inoculated (MI) control *pARK1:GUS* root lacks GUS staining. **d** Cross-section of inoculated *pArk1:GUS* and empty vector (EV) control (**e**) roots shows GUS staining exclusively in arbuscule-containing cells. Scale bar 100 μm (**a**, **b**), 50 μm (**c**), 10 μm (**d**, **e**). Representative images of three roots from five replicate plants are provided

conserved Lysine Motif (LysM) containing extracellular domain, an alpha-helical transmembrane domain (TM) and a Ser/Thr kinase domain (Supplementary Fig. 5b).

The *ARK1* gene consists of eight exons, and encodes a protein of 408 aa (Supplementary Fig. 6a). ARK1 has an N-terminal SP and a small extracellular region of only 17 amino acids, devoid of any predicted functional domains (Supplementary Fig. 6b). ARK1 contains a single TM domain and is predicted to have an active cytoplasmic Ser/Thr kinase domain (Supplementary Fig. 6b). ARK1 is thus a noncanonical putative membrane-localized RLK. Interestingly, *ARK1* is present in the genomes of mycorrhizal plant species and corresponds to *Medicago truncatula KINASE3* (*MtKin3*), but is absent from non-mycorrhizal plants species including *Arabidopsis thaliana*[48]. ARK1 has a close homolog ARK2, which is also present in *M. truncatula*, named MtKin6 that is similarly exclusively found in host plants of AM fungi[48]. Presence of a homologous ARK1 protein in the lycophyte *Selaginella moellendorfii* further suggests that ARK1 has an ancient function in AM symbiosis that arose in early land plants[48].

In silico analyses of rice LYK1 and ARK1 kinase domains, and alignment with known active kinase domains of rice and Arabidopsis CERK1, *Lotus japonicus* LjNFR1, *M. truncatula* LYK3 and the kinase-dead domain of *L. japonicus* NFR5 indicated that both rice LYK1 and ARK1 contained a predicted activation loop required for kinase activity as well as the kinase active conserved motif DFG[49] that is absent from the kinase-dead LjNFR5 (Supplementary Fig. 6c). The presence of these motifs suggests that both LYK1 and ARK1 have active kinase domains.

**Analysis of LYK1 and ARK1 function in AM symbiosis.** To determine the role LYK1 and ARK1 play during AM symbiosis, we characterized two *Tos17* retro-transposable element-generated insertion alleles[50] for each locus (Supplementary Fig. 5a, c, d and Supplementary Fig. 6a, d, e). The presence of the 4.2 kb *Tos17* element at the expected site was confirmed for each mutant by PCR (Supplementary Fig. 5c and Supplementary Fig. 6d). In the *lyk1-1* (mutant line NC0180) allele, *Tos17* was inserted into the

first exon, interrupting the extracellular domain, whereas *Tos17* disrupted the kinase domain in *lyk1-2* (mutant line NC0587, Supplementary Fig. 5a). In the *ark1-1* (mutant line NF1782) and *ark1-2* (mutant line NF4582) alleles, *Tos17* was inserted into exon 4 (upstream of the predicted kinase activation loop) and exon 5 (3′ to the predicted active kinase site), respectively (Supplementary Fig. 6a). Perturbation of transcript abundance and/or integrity for each *Tos17* insertion was confirmed (Supplementary Fig. 5d and Supplementary Fig. 6e).

For *LYK1*, the two independent insertion mutants consistently displayed wild-type colonization levels at 6 weeks post-inoculation (wpi) (Supplementary Fig. 5e), suggesting functional redundancy between LYK1 and additional LysM RLK orthologs in rice. In contrast, at 6 wpi both *ark1-1* and *ark1-2* mutants displayed significantly reduced levels of all fungal structures, although levels at 3 wpi were equivalent to wild type (Fig. 3a, b). Consistently, transcript levels of *R. irregularis* elongation factor (*RiEL*) as well as rice early and late symbiosis marker genes, *AM1* and *PT11* in *ark1-1* mutants were comparable to those of wild type at 3 wpi, but significantly reduced at 6 wpi (Supplementary Fig. 7a−c). Thus, over time the extent of fungal colonization increased in the wild type but decreased in both *ark1-1* and *ark1-2* mutants (Fig. 3a, b), suggesting reduced fungal fitness. Interestingly, we observed a striking reduction in vesicles in the mutants relative to the WT (Fig. 3b) further supporting the hypothesis that fungal vigor is compromised in the absence of functional ARK1. As arbuscules are central to fungal nourishment, it was surprising to find no evidence of abnormal arbuscule morphology in *ark1-1* and *ark1-2* mutants (Fig. 3c, d) despite the decreased fungal colonization (Fig. 3a, b). To examine whether reduced fungal colonization in the mutants was related to accelerated arbuscule turnover, we determined the distribution of arbuscule size classes. Although the overall arbuscule abundance was significantly reduced in the *ark1* mutants, arbuscule size partitioning was comparable to that of WT indicating that ARK1 is not essential for the regulation of arbuscule life span (Supplementary Fig. 8). ARK1 is thus dispensable for the initiation of AM symbiosis and for the development of arbuscules but required to maintain AM symbiosis.

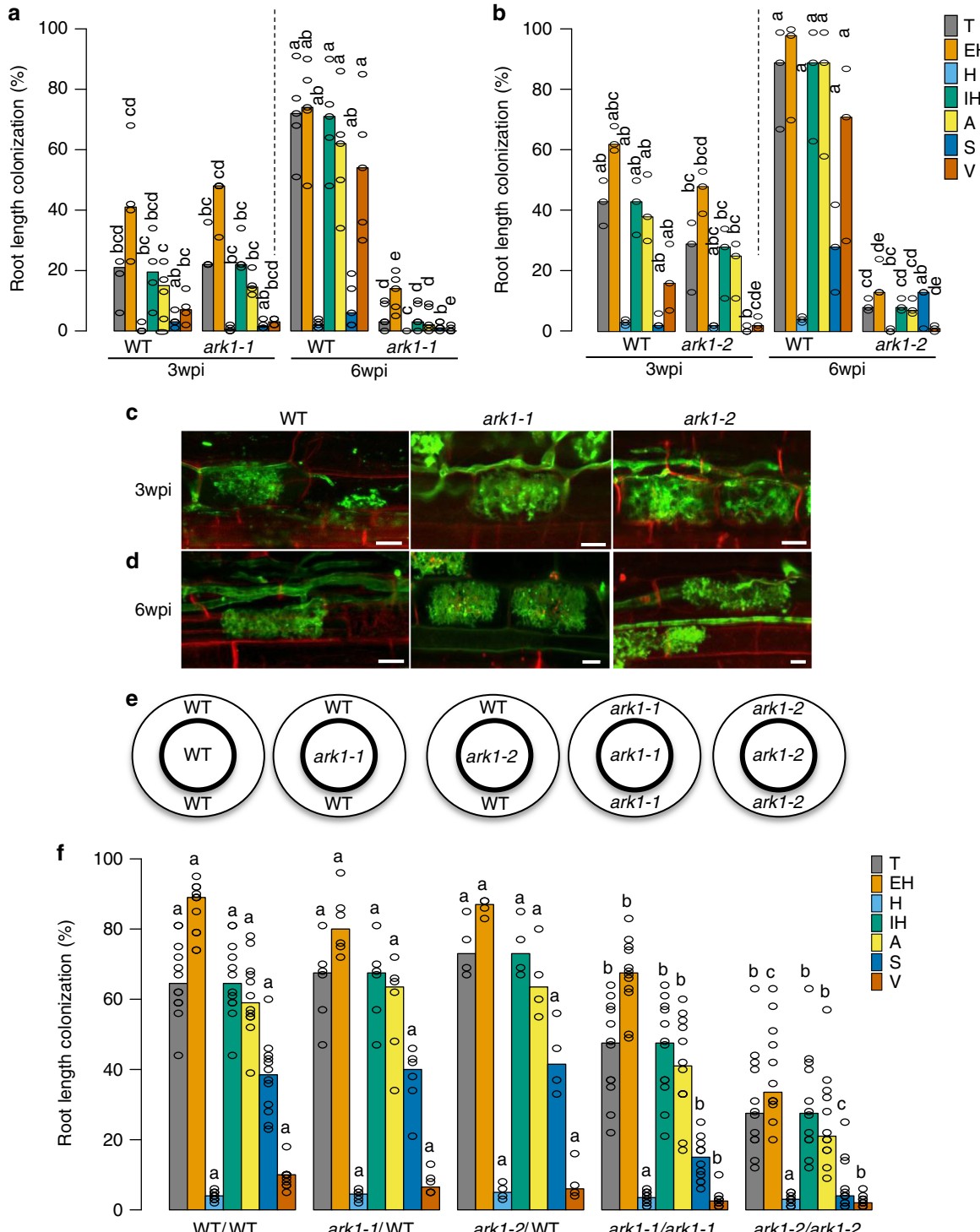

**Fig. 3** *ark1* mutants show reduced *R. irregularis* colonization after prolonged cocultivation. Percentage (%) root length colonized in wild-type and *ark1-1* (**a**) and *ark1-2* (**b**) mutant alleles at 3 weeks post inoculation (3 wpi) and 6 wpi. **c**, **d** Wheat Germ Agglutinin (WGA, green) and propidium iodide counterstained plant cell walls (red) of WT, *ark1-1*, and *ark1-2* at 3 wpi and 6 wpi, Scale bar 10 μm. Representative images of three roots from at least three plants are provided. **e** Nurse plant experimental design and **f** abundance of fungal structures (%) in nurse plant-inoculated WT, *ark1-1*, and *ark1-2* as indicated in (**e**) at 6 wpi; T total colonization, EH extra-radical hyphae, H hyphopodia, IH intra-radical hyphae, A, arbuscules, S spores, V vesicles. For statistical analysis Kruskal−Wallis tests with the Holm adjustment method were performed using the agricolae package and selecting the option for "treatment groups formation" with *P* value set at ≤0.05. Bars show median levels while the letters above each bar indicate colonization values that were not significantly different in the post hoc pairwise comparisons

AM fungal vesicles are storage structures that develop upon successful establishment of AM symbioses[13]. Due to the fatty acid heterotrophic nature of Glomeromycotina fungi[1], the presence of vesicles conceivably results from fungal uptake of plant-delivered organic carbon. We therefore asked if the reduction in vesicle formation and overall root colonization in *ark1* mutants reflected decreased fungal vigor. To this end, we examined if appropriate nourishment of the fungus by wild-type plants would rescue the *ark1* mutant phenotype by growing *R. irregularis*-inoculated mutant and wild-type plants in the same compartment (Fig. 3e, f). Remarkably, colonization levels in both *ark1-1* and *ark1-2* mutants were restored in the presence of colonized wild-type plants and thereby confirmed that ARK1 function is critical for maintaining fungal fitness.

**ARK1 localizes to the PAM.** To verify the subcellular localization of both LYK1 and ARK1, stable transgenic rice lines were generated, expressing translational LYK1-mRFP and ARK1-mRFP fusion proteins, driven by their respective native promoters. To detect fluorescently tagged proteins in inner cortex cells, deep tissue live-cell imaging was carried out using multiphoton laser scanning microscopy (MP-LSM). LYK1-mRFP fusion protein was not detectable in eight independent colonized lines. However, MP-LSM live-cell imaging of colonized ARK1-mRFP lines showed mRFP fluorescence in all lines tested. More specifically, we observed subcellular localization of ARK1-mRFP to PAM surrounding mature and collapsing arbuscule branches (Fig. 4a, b). In contrast, no RFP signal was detected in MI ARK1-mRFP or nontransformed, colonized wild-type roots (Fig. 4c, d) confirming the specificity of the RFP signal in ARK1-mRFP roots. To verify that the observed subcellular localization corresponded to a functional fusion protein, transgenic *ark1-1* lines stably expressing the ARK1-mRFP fusion protein were produced. While at 6 wpi *ark1-1* lines carrying the empty vector (*ark1-1*[ev]) control continued to show low colonization levels, significantly higher levels of colonization were observed in *ark1-1*[ARK1-mRFP] compared to *ark1-1*[ev] control lines (Fig. 4e), demonstrating at least partial functionality of the ARK1-mRFP fusion protein.

During arbuscule development the separation between the PAM and closely associated subcellular compartments cannot be discriminated by MP-LSM. To increase resolution and verify PAM-localization of ARK1, immuno-gold labeling transmission electron microscopy (TEM) was performed. The pattern of immuno-gold particle distribution confirmed that ARK1-mRFP accumulated only at the PAM (Fig. 4f, g). Specificity of the anti-RFP antibody was confirmed by the lack of RFP signal on PAM from OsPT11-eGFP-expressing root tissue (Fig. 4h). To our knowledge, no other AM symbiosis-induced RLK has been shown to localize to the PAM prior to ARK1.

## Discussion

The plasma membrane proteome is inherently dynamic as each cell type needs to continuously adjust its unique requirements to the changing environment. Within the arbusculated cell, the ephemeral PAM provides a substantial surface area at the intracellular plant–fungal interface. Unique transport proteins reside within the PAM, thereby defining a central role of the PAM in symbiotic nutrient uptake. Coordination of the cellular activities of both symbionts conceivably involves signaling at the PAM; however, the mechanism is currently elusive. Among the candidates for plant–fungal communication at the PAM, RLKs containing chitin-binding LysM motifs in their extracellular domain have been extensively studied in the context of AM fungal detection (reviewed in ref. [51]). Yet, while most studies have focused on the perception of microbial signals in the rhizosphere,

little information is available on RLKs being required for the physical part of the interaction. In *Parasponia andersonii* RNAi-mediated downregulation of the LysM-RLK Nod Factor Perception (PaNFP) revealed a critical role for arbuscule development[52]. However, the simultaneous silencing of multiple homologous LysM-RLKs may have accounted for the observed phenotype, and the subcellular localization of PaNFP was not determined. Our study identified a LysM-RLK, LYK1, which accumulated in mycorrhizal roots of maize and rice. On the basis of the restricted promoter activity in arbuscule-containing cells, we infer that the protein accumulates specifically in arbusculated cells, however, most probably at levels too low to be detected by confocal microscopy. This is consistent with previous transcriptomic analyses where lower levels of *LYK1* transcripts were found upon AM fungal colonization compared to that of *ARK1* and *PT11* [43,53]. At the same time this highlights the sensitivity of our newly established proteomics workflow and the efficacy of the LCM enrichment strategies. The functional relevance of LYK1 for AM symbiosis development remains, however, unknown. Although other LysM-RLKs were not identified in our proteomics/LCM-transcriptomic comparison, genetic redundancy is possible as our analysis permitted only the recovery of uniformly highly expressed genes/proteins.

Importantly, our approach identified a second putative Ser/Thr RLK, the PAM-specific ARK1 that is essential for sustained AM symbiosis, thereby providing evidence for signaling at the PAM. Mutation of *ARK1* led to a drastic reduction in vesicle formation, indicating that the fungus was compromised in completing its life cycle, a phenotype that became apparent only after prolonged plant fungal cocultivation. The molecular processes that underpin the production of vesicles have received little attention. However, as storage structures, vesicles contain large amounts of lipids and a functional link between arbuscule development, plant-delivery of lipids and vesicle formation has recently been established in legumes[5]. Although vesicle biogenesis has not been explicitly studied, the requirement for induced plant fatty acid production in arbusculated cells to nourish the fungus was consistently correlated with vesicle abundance[3,22]. Common to these studies is the inability of the fungus to produce fully developed arbuscules in plant mutants defective in lipid biosynthesis or delivery. The stunted arbuscule phenotype has been proposed to result from an insufficient supply of plant fatty acids to support arbuscule membrane biogenesis[2–5] (reviewed in refs. [54,55]). Surprisingly, although *ark1* mutants phenocopied low vesicle abundance and reduced overall colonization, arbuscule development per se was not compromised, indicating that ARK1 function conditions processes that take place after arbuscules have formed. Upon initial colonization, arbuscules are generated rapidly before vesicles emerge, likely being fueled by resources provided by spores and by intra-radical hyphae[14]. In rice *ark1* mutants, the low vesicle number indicates that the symbiosis fails to mature, and that the fungus is not "keeping up" with host root growth as illustrated by the increasingly lower total root length colonization at later time points post inoculation. A consistent observation was made in mutants of the homologous *M. truncatula KIN3*, where total root colonization levels dropped with extended cultivation, although a more detailed phenotypic characterization was not included in the study[48]. The recovery of vesicle and spore production in *ark1* mutants cocultivated with wild-type plants points towards a requirement of ARK1 for maintained fungal vigor. In our view, this is an intriguing finding because it links cellular processes occurring post arbuscule development with fungal fitness, thereby defining a previously unknown control step in AM symbiosis. Since the ARK1 kinase domain has the hallmarks of an active kinase, it is likely that ARK1-mediated signaling is key to ensuring continuous intra-radical fungal proliferation.

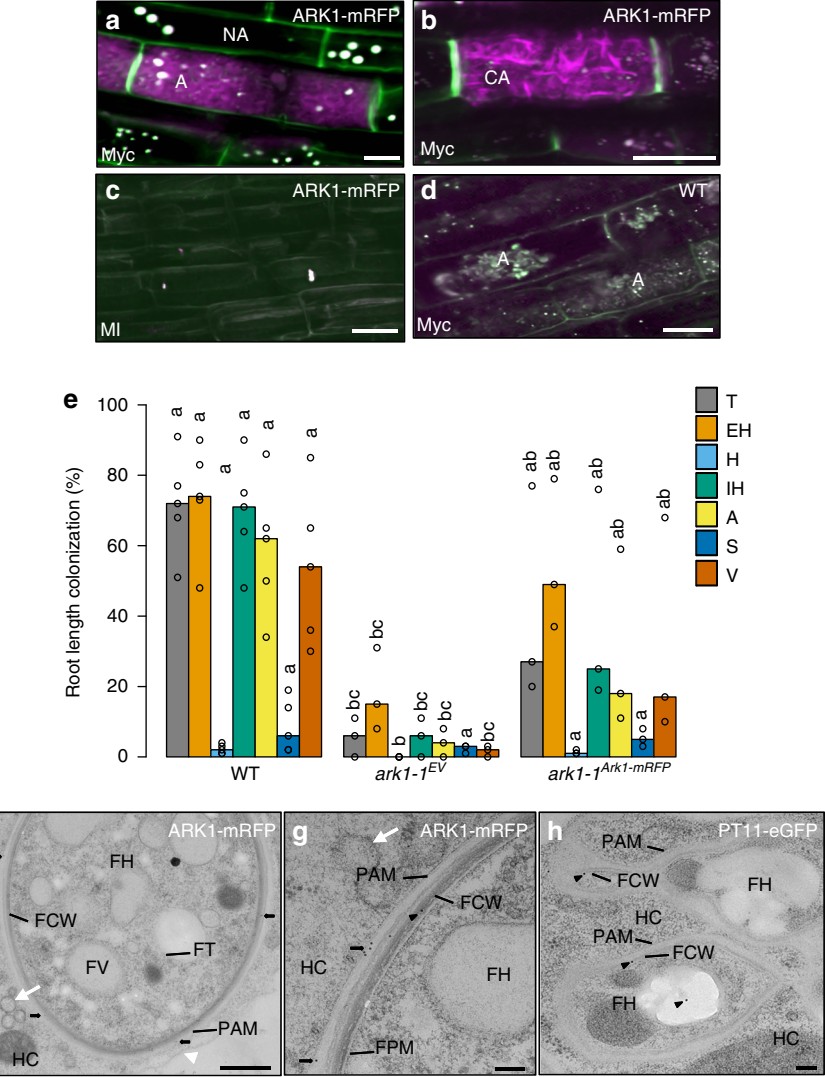

**Fig. 4** ARK-mRFP localizes to the peri-arbuscular membrane. Multiphoton laser scanning microscopy (**a**−**d**) shows subcellular localization of **a** mRFP (magenta) signal labels PAM around arbuscule fine branches (A) at 20 days post-inoculation (dpi). Auto fluorescence from plant cell walls and autofluorescent bodies are false colored in green and white, respectively. NA nonarbusculated cell. Scale bar 10 μm, z-stack 10 μm. **b** ARK1-mRFP localizes to PAM surrounding collapsing arbuscular (CA) fine hyphae. Scale bar 30 μm, z-stack 20 μm. **c** Mock-inoculated (MI) ARK1-mRFP-expressing roots and **d** nontransformed colonized WT lack RFP signal; **c**, **d** Scale bar 15 μm, z-stack 10 μm. Representative images of three roots from at least three plants are provided. **e** Quantification of *R. irregularis* root colonization levels of the *ark1-1* mutant containing the empty vector (EV) control or *pARK::ARK-mRFP*. T total colonization, EH extra-radical hyphae, H hyphopodia, IH intra-radical hyphae, A arbuscules, S spores, V vesicles. For statistical analysis Kruskal−Wallis tests with the Holm adjustment method were performed using the agricolae package and selecting the option for "treatment groups formation" with *P* value set at ≤0.05. Bars show median levels while the letters above each bar indicate colonization values that were not significantly different in the post hoc pairwise comparisons. **f**−**h** Transmission electron micrograph (TEM) shows immuno-gold labeling (IGL) of ARK1-mRFP to the PAM (**f**, **g**) using an RFP antibody and secondary antibody conjugated to 10-nm immuno-gold particles (black arrows) and not to ER (white arrow) or host tonoplast (white arrowhead). Black arrowheads show nonspecific labeling to the fungal cell wall (FCW). Scale bar 500 nm (**f**) and 100 nm. Representative images from arbuscule-containing cells (*n* > 4) of independently sampled roots derived from two different experiments (**g**). **h** PT11-eGFP control roots cross-reacted with RFP antibodies, showing nonspecific labeling (black arrowheads) of the FCW. Scale bar 100 nm. HC host cytosol, FH fungal hypha, FT fungal tonoplast, FPM fungal plasma membrane

Interestingly, ARK1 lacks a discernible extracellular region, which draws into question the mechanism of ARK1 function and points to a coreceptor model for ARK1. Together, our study provides insight into PAM-resident RLK signaling in AM symbiosis.

## Methods
**Plant cultivation and inoculation with *R. irregularis*.** *Zea mays* inbred line B73 and *Oryza sativa* ssp. japonica cv. Nipponbare rice seeds were used throughout this study. Spores of *R. irregularis* were purchased (Premiertech, Rivière-du-Loup,

Canada). Maize and rice seeds were surface-sterilized with 3.5% sodium hypochloride solution for 15 min, washed extensively with sterile water and pregerminated in sterile water for 5 days at 25 °C in the dark. Maize and rice seedlings were inoculated upon planting with 800 or 250 *R. irregularis* spores, respectively. For TEM and live-cell imaging, to ensure rapid colonization within 20 days post inoculation (dpi), pregerminated rice seedlings were grown in the presence of a nurse plant in 4.5 cm petri dishes. For physiological nurse plant experiments, plants were grown in pots, where tester plants were surrounded by six donor plants as shown in Fig. 3e. Plants were grown in a growth chamber with a 12 h/12 h day/night cycle at 28 °C/22 °C and 60% humidity. Plants were fertilized

every second day with full (maize) or half (rice) Hoagland solution, containing 100 $\mu$M of $KH_2PO_4$ or 25 $\mu$M $KH_2PO_4$ respectively. Plants were harvested 7 wpi and stained with trypan blue to quantify fungal colonization. For trypan blue staining root samples were incubated in 10% KOH for 30 min at 96 °C. Following three washes with distilled water, samples were incubated in 0.3 M HCl for 30 min at room temperature, followed by boiling at 96 °C for 8 min in a 0.1% w/v trypan blue staining solution in a 2:1:1 mixture of lactic acid, glycerol, and distilled water. Roots were de-stained with a 1:1 solution of glycerol and 0.3 M HCl. Total root length colonization referring to total fungus or to specific structures was scored microscopically at 100 random points per root sample. For reliable representation of root length colonization by specific mycorrhizal structures, all structures present at one random point were separately recorded. For statistical analysis Kruskal−Wallis tests with the Holm adjustment method were performed using the agricolae package and selecting the option for "treatment groups formation" with $P$ value set at ≤0.05 [56]. Trypan blue-stained roots were photographed using an Olympus Bx43 microscope and QCapture Pro image analysis software. For live-cell imaging or TEM plants were grown in a nurse plant system in 45 mm petri dishes and seedlings imaged at 21 dpi.

**Protein extraction and plasma membrane enrichment**. For protein extraction a total of 40 g root fresh weight was used both for rice and maize. Roots were ground using a laboratory blender and subsequently homogenized in ice-cold extraction buffer (25 mM Hepes-KOH pH 7.2, 200 mM Sorbitol, 1 mM dithiothreitol (DTT) and 2 mM phenylmethylsulfonyl fluoride (PMSF)). Samples were filtered through two layers of Miracloth (Calbiochem, San Diego, CA, USA). The samples were subjected to ultracentrifugation at $100,000 \times g$ for 45 min at 4 °C. The microsomal fractions (MFs), were resuspended in STED 10 (10% sucrose w/w, 0.5 M Tris-HCl pH 7.5, 0.5 M ethylene-diamine-tetra acetic acid (EDTA) and 0.5 M DTT) and then subjected to overnight centrifugation at 100,000 g at 4 °C in a continuous sucrose gradient. A total of 14 fractions were collected from each sample, and the sucrose concentration was measured. The fractions between sucrose concentration of 33–38%, enriched in PM[57,58] were collected, then washed in 20 mM Tris/Hepes pH 7.2 and 1 mM $MgCl_2$ and centrifuged at $100,000 \times g$ at 4 °C for 60 min.

**Protein solubilization and digestion**. Protein fractions enriched for PM (PMPF) were prepared for mass spectrometry using the filter-aided sample preparation (FASP) method[59]. Briefly, PMPF were resuspended in SDS-Lysis buffer (4% w/v Sodium dodecyl sulfate (SDS), 100 mM Tris-HCl pH 7.6 and 100 mM DTT) using the minimum volume of buffer of ~400 μl for solubilization. The solutions were sonicated for 5 min at 50% cycle time and then heated for 5 min in a boiling water bath. Protein concentrates were diluted tenfold using 8 M urea in 0.1 M Tris-HCl, pH 8.5 (UA buffer), loaded onto the 30k microcon filter units (Millipore, Zug, Switzerland) and centrifuged at $14,000 \times g$ for 10 min at 25 °C. This step was repeated as required until all lysates were loaded. Then 50 μl of 40 mM iodoacetamide in UA buffer was added to the filters and the samples were incubated in the dark for 10 min. Filters were washed twice with 250 μl of UA buffer followed by two washes with 250 μl of 50 mM $NH_4HCO_3$. Protein digestions were conducted overnight at 37 °C with 1:100 trypsin to sample (Promega, Dübendorf, Switzerland). After digestion, the liberated peptides were collected by centrifugation at $11,000 \times g$ for 20 min. The concentration was estimated by nanodrop UV spectrophotometry, assuming that a 0.1% (1 mg ml$^{-1}$) solution of vertebrate proteins has an absorbance of 1.1 at 280 nm. The used concentrations were 8 μg μl$^{-1}$ for PMPFs. All samples were stored at −20 °C until further processing.

**Establishment of PAM-enriched membrane proteome workflow**. Our workflow (Supplementary Fig. 2c) commenced with the traditional continuous sucrose density centrifugation approach[60], which was preferred over aqueous two-phase partitioning to assure recovery of PAM within enriched membrane fractions, since electrostatic and hydrophobic properties of the PAM are elusive. In line with published reports, PM fractions were enriched between sucrose concentrations of 33–38%[57] as monitored with a PM-specific anti-H$^+$-ATPase[32] antibody (catalog no. AS07260, Agrisera AB, https://www.agrisera.com/) (Supplementary Fig. 2d)[61]. The complete solubilization of the membrane proteome by FASP[59] increased the detection of underrepresented PM intrinsic proteins. We added equal volumes of PM-enriched MFs to the filters, conducted a trypsin digestion overnight and the population of peptides was collected by centrifugation. To lower sample complexity and increase the resolution of the peptides, strong cation exchange (SCX) chromatography was included as a prefractionation step[62]. The collected peptide mixture from each sample was brought to an equal concentration and loaded onto the SCX column. The resulting 27 fractions were pooled into eight master fractions for LC-MS analysis. Mass spectrometry analysis of peptides was performed on a linear ion trap—Orbitrap instrument (LTQ Orbitrap Velos), which alternatively provides high acquisition rates and sensitivity, or high mass accuracy[63]. To correctly interpret MS spectra we applied the target-decoy strategy[64], which permits identification of peptides and proteins at a given false discovery rate[65].

**Peptide separation and mass spectrometry analysis**. Peptides were loaded on a 2.1 mm inner diameter × 200 mm long SCX column (column material polysulphoethyl A, 5 μm particle size, 200 Å pore size, Poly-LC, www.polylc.com) and

eluted at a flow rate of 0.2 ml min$^{-1}$. The 27 fractions obtained were pooled to eight master fractions according to the SCX chromatogram, and desalted using ZipTips C$_{18}$ (Millipore, Zug, Switzerland). The samples were analyzed on an LTQ Orbitrap Velos mass spectrometer (Thermo Fisher Scientific, www.thermoscientific.com) coupled to an Eksigent Nano HPLC system (Eksigent Technologies, www.eksigent.com). The solvent composition of buffer A was 0.1% formic acid in $H_2O$, and buffer B contained 0.1% formic acid in acetonitrile. Samples were dissolved in 3% acetonitrile and 0.1% formic acid. Peptides were loaded onto a self-made tip column (75 μm inner diameter × 80 mm long) packed with reverse-phase C18 material (AQ, particle size 3 μm, pore size 200 Å; Bischoff GmbH, www.bischoff-chrom.com) and eluted at a flow rate of 250 nl min$^{-1}$. The following LC gradient was applied: 0–5 min 2% B; 95 min 50% B; 98 min 70% B; 100 min 100% B; 105 min 100% B; 108 min 2% B; 140 min 2% B. Mass spectra were acquired in the m/z range 300–1700 in the Orbitrap mass analyzer at a resolution of 30,000 at m/z 400. MS/MS spectra were recorded in a data-dependent manner for the 20 most intense signals in the ion trap. Precursor masses already selected for MS/MS acquisition were excluded from further selection for 45 s, and the exclusion window was set to 10 ppm. FT preview mode was enabled, and the poly-dimethylcyclosiloxane background ion at 445.120025 was used for internal calibration (lock mass).

**Protein identification and data processing**. The MS and MS/MS data were converted into Mascot generic files (mgf) with Mascot Distiller 2.4 using automated and rule-based tool[66]. The Mascot search engine (Matrix Science, version 2.4) was used for database searches. MS and MS/MS data were searched against a target-decoy database containing forward and reversed protein sequences from *Zea mays* (www.maizesequence.org) and *R. irregularis* (www.uniprot.com) or from *Oryzae sativa* (http://rice.plantbiology.msu.edu/). Furthermore, a set of 260 known protein contaminants such as keratin and trypsin was added. The following search parameters were applied: (i) trypsin was used as the protein-digesting enzyme, and up to two missed cleavages were tolerated, (ii) carbamidomethylation of cysteine was specified as a fixed modification, (iii) oxidation of methionine and pyro-Glu for N-term Gln were selected as variable modifications. Searches were performed with a parent-ion mass tolerance of ±10 ppm and a fragment-ion mass tolerance of ±0.6 Da.

Scaffold version 4.0 (Proteome Software Inc, Portland, Oregon) was used to validate and quantify MS/MS-based peptide and protein identifications. We filtered the data with Scaffold in order to keep the FDR at the protein level below 1% and at least two unique peptides were required for protein identification. Only proteins with a total of at least four peptide spectrum matches were considered quantified. The quantitative values are based on Scaffold software algorithm. Proteins that were identified using the same set of peptides and were not differentiated by the MS/MS analysis were grouped to protein clusters to satisfy the principles of parsimony. Differential expression values are expressed in log$_2$ scale.

**Rice versus maize protein Blast search**. To compare rice and maize protein identifications we carried out a blast search using BLASTP 2.3.0+ with default parameters on maize and rice protein databases. The best single hit was taken. The mass spectrometry proteomics data have been deposited to the ProteomeXchange Consortium (http://proteomecentral.proteomexchange.org) via the PRIDE partner repository[67] with the dataset identifier PXD002575 for maize and PXD010313 for rice.

**Affymetrix Genechip$^{TM}$ rice genome array hybridization**. Total RNA (50–100 ng) was subjected to amplification by the WT-Ovation$^{TM}$ One-Direct amplification system (NuGEN, http://www.nugeninc.com). This technology was designed for direct amplification of RNA in picogram quantities in cell lysates. For expression analysis, up to 3 μg of amplified biotin-labeled cDNA was hybridized to Affymetrix GeneChip rice genome arrays (Affymetrix, http://www.affymetrix.com/). Hybridization, washing steps, staining, and scanning were performed according to the manufacturer's instructions.

**Gene expression analysis**. RNA was extracted from 100 mg ground root tissue by the standard Trizol protocol according to the manufacturer's instructions (Fisher Scientific UK Ltd, https://www.fishersci.co.uk). Total RNA was DNase I-treated according to the provided protocol (Sigma-Aldrich, https://www.sigmaaldrich.com). Prior to first-strand cDNA synthesis, the absence of genomic DNA in the RNA samples was confirmed by PCR. First-strand cDNA synthesis was carried out following the manufacturer's instructions using SuperScript II Reverse Transcriptase (Invitrogen, Fisher Scientific UK Ltd, https://www.fishersci.co.uk). Primer design for real-time RT-PCR was carried out with the NCBI design tool Primer-BLAST. PCR-amplification efficiencies were established on twofold serial template dilutions. For qRT-PCR-based gene expression analysis, relative transcript levels were normalized to the geometric mean of amplification of three near-constitutively expressed genes: *CYCLOPHILIN2* (LOC_Os02g02890), *GAPDH* (LOC_Os08g03290), and *POLYUBIQUITIN* (LOC_Os06g46770). Normalized expression values were displayed as a function of *CYCLOPHILIN2* expression.

**Laser capture microdissection**. Fresh roots of mycorrhizal and non-mycorrhizal rice plants were collected at 7 wpi, cut into 3–5 mm pieces and immediately acetone-fixed and embedded employing a microwave-enhanced paraffin embedding. Root sections were arranged in parallel on UV-treated poly-L-lysine slides. A laser microbeam system (P.A.L.M. Microlaser Technologies, http://www.zeiss.de/micro-dissection) was utilized for microdissection of cortical root cells. For cutting, the following parameters were selected using P.A.L.M. Robosoftware 2003 (http://www.zeiss.de/microdissection): auto-LPC focus 50, with energy of 95 and a speed of 100. Approximately 2000 cortical cells were isolated as combined cell units. Isolated specimens were catapulted onto the adhesive surface on the inside of the lid of a 0.5 ml reaction tube (Carl Zeiss, http://www.zeiss.com/), collected by centrifugation (11,000 × $g$ for 30 s) and immediately subjected to 350 μl RLT buffer (Qiagen, http://www.qiagen.com/, 40 units), 1 μl RNaseOUT (Invitrogen, http://www.invitrogen.com/) and 3.5 μl β-mercaptoethanol (Sigma-Aldrich, http://www.sigmaaldrich.com/). The sample was vigorously mixed and incubated at 56 °C for 5 min to facilitate cell-wall disruption. At this stage, the lysate was either stored at −80 °C or further processed for RNA isolation. Total RNA of each cell population was extracted using an RNeasy micro kit (Qiagen) according to the manufacturer's instructions. The RNA concentration was determined using NanoDrop ND-1000 (NanoDrop Products, http://www.nanodrop.com). The RNA quality was assessed by determining the $A_{260/280}$ ratio. In addition, the integrity of RNA molecules was determined by utilization of an RNA 6000 Pico LabChip Kit and Bioanalyzer 2100 (Agilent Technologies, http://www.agilent.com).

**LCM data analysis**. The CEL files were imported into R/Bioconductor[68] (R Core, http://www.R-project.org/, Vienna, Austria) using the affy package[69] and normalized with the Robust Microarray Average algorithm[70]. Annotation was obtained from Ensembl Plants (http://plants.ensembl.org/index.html). Affymetrix probes that did not map to any Ensembl Gene ID were excluded, and if more than one probe mapped to the same Gene ID, only the one with the highest Inter Quartile Range (across all samples) was kept. Differential expression analysis was performed in the limma package[71] using lmFit and eBayes functions to make pairwise contrasts between the three treatments. To select differentially expressed genes for comparison with the proteomics dataset, the decideTests function was used, with the "nestedF" method, adjusted for multiple testing using the Benjamini−Hochberg adjustment, setting thresholds of adj-$P = 0.15$ and $\log_2 FC > 1$. A set of 428 transcripts was identified to differ in at least one contrast. Running the decideTests function without specifying the $\log_2 FC$ threshold identified eight additional downregulated transcripts (not shown); the FC threshold was retained for the subsequent analysis. Running the decideTests function using the "global" method resulted in a similar, if slightly larger transcript set (566 transcripts); the "nestedF" method was retained for the identification of the final transcript set.

**Constructs for Promoter-GUS and fluorescent protein tag**. The promoter regions of ARK1 (1953-bp; LOC_Os11g26140) and of LYK1 (1471-bp; LOC_Os01g36550) were cloned into the pENTR/D-TOPO vector (Supplementary Table 6) and transferred into a Gateway-compatible pHGWFS7.0 vector containing an enhanced green fluorescent protein-GUS fusion reporter gene[72]. For the translational fusion constructs, ARK1 (4983-bp) and LYK1 (4455-bp) genomic regions were cloned into pENTR/D-TOPO vector (Supplementary Table 6) and subsequently moved into the Gateway-compatible pGWB553 vector, containing a monomeric red fluorescent protein[73].

**Rice transformation**. Mature dehusked seeds of rice cultivar Nipponbare were surface sterilized, thoroughly rinsed with sterile deionized water and air dried in a flowbench. Seeds were plated, after removal of the embryo axes, on N6DT medium essentially as previously described[74] supplemented with timentin (150 mg l$^{-1}$). Plates were sealed, cultured in the dark at 28 °C for 17 days and subcultured 4 days prior to transformation. Callus pieces were inoculated with an overnight culture of Agrobacterium tumefaciens strain EHA105, containing the appropriate construct, resuspended in AAM medium (0.2–0.3 OD$_{600}$). After 5 min, the Agrobacterium suspension was removed and the inoculated callus transferred to sterile filter paper in a 9 cm petri dish. Plates were sealed with Parafilm and cocultivated for 3 days at 25 °C/23 °C in the dark, followed by transfer to N6DT with 50 mg l$^{-1}$ hygromycin. Subsequent tissue culture media were essentially as described by Sallaud et al.[75] except for the substitution of NB basic with MS salts and of cefotaxime/vanco-mycin with timentin, as above. Callus pieces were transferred to fresh medium at 10-day intervals with shoots removed for rooting after approximately 4–5 weeks. Rooting was undertaken on HF medium as described[74] with timentin and 25 mg l$^{-1}$ hygromycin. Genomic DNA was isolated from rooted plantlets[76] and plants were confirmed as transformed by multiplex PCR using primers for the hpt selectable marker gene (HygF-UP and HygR-UP) and a cyclophillin endogenous control gene (Cyp2-442F and Cyp2-859R).

**Live-cell imaging**. Live-cell imaging was carried out using an MP-LSM (TriM Scope II, LaVision Bio Tec) containing a dual-line femtosecond laser tunable to 680–1300 nm and a fixed 1040 nm laser (Insight DeepSee, Spectra-Physics). RFP fluorescence was detected by exciting with the 1040 nm laser line and emission collected above 560 nm using a long-pass filter. Autofluorescence was detected by exciting roots at 850 nm. Fluorescence corresponding to RFP was obtained by subtracting fluorescent emissions obtained from exciting at 1040 nm (RFP) and 850 nm (autofluorescence). A minimum of four roots were imaged per plant. Roots remained attached to the whole plant immersed in water and were imaged using an Olympus 25 × 1.05 NA water dipping objective. Images were denoised using denoising software[77]. Images were analyzed using IMARISx64 8.1.2 image analysis software (Bitplane, http://www.bitplane.com) and Fiji[78].

**Histochemical GUS and wheat germ agglutinin staining**. Histochemical GUS staining was performed using the slightly modified protocol from ref. [79]. Fresh root samples were covered with GUS staining solution containing 100 mM sodium phosphate buffer (pH 7.0), 10 mM sodium EDTA, 0.5 mM potassium ferrocyanide, 0.5 mM potassium ferricyanide, 0.10% Triton X-100 and 1 mM 5- bromo-4-chloro-3-indolyl-β–D-glucuronic acid (X-gluc; Melford). Samples were vacuum filtrated three times for 5 min each and incubated for 48 h at 37 °C and the reaction terminated by placing roots in 50% ethanol. Roots were imaged following clearing in a solution of 20% KOH for 2–3 days. For wheat germ agglutinin (WGA) staining, root samples were collected in 50% ethanol and incubated overnight followed by 2–3 days in 20% KOH. After rinsing in distilled water, roots were incubated in 0.1 M HCl for 2 h followed by incubation in 0.2 μg ml$^{-1}$ WGA-AlexaFluor488 (catalog no. W11261, Fisher Scientific UK Ltd, https://www.fishersci.co.uk) staining solution in 1× phosphate-buffered saline (PBS) for 2–3 days. Prior to imaging plant cell walls were counter-stained for 3 min in a 10 μg ml$^{-1}$ solution of propidium iodide in water (catalog no. P3566, Fisher Scientific UK Ltd, https://www.fishersci.co.uk).

**Transmission electron microscopy**. For high-pressure freezing, rice sectors with ARK-RFP expression were dissected and submerged in freezing medium (200 mM sucrose, 10 mM trehalose and 10 mM Tris buffer, pH 6.6)[80]. Root samples were immediately transferred into aluminum planchettes (type 214 and 242; Wohlwend GmbH, http://www.wohlwend-hpf.ch) and frozen using a Baltec HPM010 high-pressure freezer (ABRA Fluid AG, http://www.abra-fluid.com). Freeze-substitution was carried out according to the method developed by Hillmer et al.[80] in the absence of osmium. Briefly, freeze-substitution was carried out in 0.4% uranyl acetate in acetone (stock solution 20% uranyl acetate in methanol) in molded Alu-caps (Diagonal catalog no. 3621313, www.diagonal.de), which were placed inside Alu-weighing dishes (VWR catalog no. 611-1362, www.vwr.com) as outer containers for polymerization. Freeze-substitution was done in a Leica AFS2 base unit (Leica, https://www.leica-microsystems.com) equilibrated to −85 °C for 16 h followed by a 5 h linear warm up to −60 °C and a 1 h wash with dry ethanol at −60 °C, at which point root samples were released from planchettes using precooled needles. Infiltration with resin was done in a stepwise manner lasting 1 h each at −60 °C from 30%−50%−70% Lowicryl HM20 in ethanol to 100% Lowicryl HM20. Polymerization was carried out in the Leica AFS2 unit under ultra-violet (UV) maintaining a temperature of −60 °C for 24 h followed by a gradual increase to room temperature at the end of the second day. For immuno-gold labeling ultrathin (70 nm) sections were cut on a Leica Ultracut E microtome (Leica Microsystems, www.leicabiosystems.com) and labeling was carried out for 1 h using a polyclonal anti-RFP (Clontech; Living Colors DsRed polyclonal rabbit antibodies, catalog no. 632496 https://www.takarabio.com) or anti-GFP antibodies at a dilution of 1:800 or 1:1000 in PBS (containing 1% BSA). Anti-GFP antiserum was produced from recombinant GFP (Axxora Product. No. VC-MB-0752-C100), which was used to immunize rabbits according to standard protocols by Agrisera. Following incubation with primary antiserum sections were incubated with 10-nm gold-coupled secondary antibodies (BioCell GAR10, https://www.bbisolutions.com) at a dilution of 1:50 in PBS supplemented with 1% BSA. Sections were observed in a Tecnai G2 80–200 kV TEM (https://www.fei.com) operating at 80 kV.

## Data availability

Peptide sequences have been uploaded to the public database PRIDE via the ProteomeXchange website with the dataset identifier PXD002575 for maize (Project Webpage: http://www.ebi.ac.uk/pride/archive/projects/PXD002575; FTP Download: ftp://ftp.pride.ebi.ac.uk/pride/data/archive/2018/09/PXD002575) and PXD010313 for rice (Project Webpage: http://www.ebi.ac.uk/pride/archive/projects/PXD010313; FTP Download: ftp://ftp.pride.ebi.ac.uk/pride/data/archive/2018/09/PXD010313).

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

## Acknowledgements

We thank Anne Bates and Steffi Gold for technical assistance, Yoshihiro Kobae for kindly providing PT11-GFP transgenic rice lines and Jonathan Schnabel for writing the R script used to depict the *R. irregularis* colonization data. R.R. was supported by the Marie Curie FP7-PEOPLE-2013-IEF Grant number 629887. Work in the laboratory of U.P. was supported by funds from the University of Lausanne (M.C.), the Gatsby Charitable Foundation RG60824, the Isaac Newton Trust RG74108, and the BBSRC grant BB/N008723/1. Work in the laboratory of E.J.W. was funded by the NIAB Trust.

## Author contributions

R.R., M.C., J.G., E.J.W., E.M., and U.P. conceptualized the project; R.R., M.C., H.M., P.G., D.H., F.W., S.-Y.Y., S.B. and U.P. carried out the experiments; M.C. and R.W. conducted the computational analysis of genomics datasets; R.R., M.C., and R.S. performed the statistical analyses; A.M. generated rice mutant lines; R.R. and U.P wrote the manuscript.

## Additional information

**Competing interests:** The authors declare no competing interests.

