## [Peer Review File · Nature Communications]

Reviewers' Comments:

Reviewer #1:

Remarks to the Author:

The manuscript identified a new receptor-like kinase that is located on the periarbuscular membrane of mycorrhizal roots in rice and plays a role in the long term maintenance of arbuscules. This is an important finding in the field because no genes with similar functions have so far been identified. It was particularly interesting to find that co-cultivation of the fungal partner on wild type and mutant plants at the same time could restore the ability to maintain the symbiosis on the mutant, and this will open up interesting future studies to identify potential signals or nutrients that are provided by the host to maintain symbiosis.

The authors first used a proteomics approach to identify proteins located on the periarbuscular membrane. This was followed by a functional approach using rice mutants, combined with microscopy to localize the protein.

I found the manuscript very interesting to read. However, some of the documentation was confusing and some missing, as outlined below. Some additional controls for the microscopy also need to be included.

General Comments:

Somewhere in the results (supplement) the authors need to show the details of the protein identification, i.e. a list of proteins with the number of peptides that each protein was identified with, the peptide sequences, the number of mis-cleavages and allowed modifications and the mass accuracy.

A lot more proteins were identified to be enhanced in mycorrhizal roots compared to mock inoculated roots in the whole root approach compared to micro-dissected cells. Only 43 transcripts were found to be differentially accumulated in arbuscule-containing cells (line 222), while in the proteomic approach, 1180 proteins were stated to be exclusively found in colonized roots (line 199). Could you discuss the likely function/location of the remaining differentially accumulated proteins?

A lot more rice proteins were identified than maize proteins. Could you discuss the reasons for this?

Overall there was a surprisingly small overlap between proteins and transcripts detected in mycorrhizal roots/arbuscule-containing cells. The authors did not really speculate on this but something should be included in the discussion; i.e. what are all the other almost 500 / 900 proteins that were detected in the proteome analysis of microsomal enriched mycorrhized roots compared to the arbuscule containing cells?

Specific comments:

Line 180: Previous reports are mentioned here but not referenced. Please include a reference to studies that this one is compared to.

Line 304: Please check the end of this sentence (delete 'at?') and define 'wpi' at the first mention: 'at six wpi both ark1-1 and ark1-2 mutants displayed significantly reduced levels of all fungal structures, whereas at three weeks post inoculation (wpi) both ark1 mutant lines revealed equivalent levels of most fungal structures relative to the wild-type at'

Line 208: 'The relatively large number of 524 and 1011 proteins not shared between maize and rice (Fig. 2c) could represent species-specific proteins in addition to technical noise.' What do the authors refer to with 'technical noise'? Have other proteomics studies found large numbers of species-specific proteins? Could the reason include the naming of proteins in different databases (i.e. proteins with the same functions could have different names and not be matched).

Line 242: 'LYK1-mRFP fusion protein was not detectable in 8 independent colonized lines (data not shown).' What could be the reason?

Line 333: When saying 'by growing *R. irregularis* inoculated mutant and wild-type plants in the same compartment', does this mean that the fungus can form a symbiosis with both WT and mutant plants at the same time by growing hyphae concurrently into each root?

Line 363: OsPT11-eGFP was used as a control for RFP antibody specificity, why was this construct chosen?

Line 444: 'Wild-type nurse plant inoculation was performed in 4.5cm petri dishes to achieve rapid colonization within 20 days post inoculation' Were maize or rice plants indeed grown in 4.5 cm petri dishes for this experiment? It was hard to visualize them fitting into those plates without restricting the root growth.

Line 489: 'The used concentrations were 8 μ g'. 8 μ g is not a concentration but an amount.

Line 500: Please detail the conditions used for HPLC and coupled mass spectrometry in the Orbitrap mass analyzer.

Line 547: this should probably read '-80 degrees'.

Figures:

Figure 1 was very helpful!

Figure 2: I did not completely understand this figure: In panels a and b, what are the numbers in the circles, and why are there two rows for each, e.g. Plant proteins: 776 – 15.6%, and underneath: 91 – 1.8%? Please explain this in the legend.

Figure 3: It appears from the white arrow indicating arbuscules that the GUS staining is not specific for cells containing arbuscules. Is this correct, or are not all arbuscule-containing cells marked with an arrow? The roots need to be counterstained with ink (or similar) to indicate GUS staining at the same time as arbuscules.

Figure 5f: I assume that the significant differences are found between the first three bars and the last two bars, is this correct? It might be easier to indicate this with letters, e.g. the label the first three bars with 'a' and the last two with 'b'.

Figure 6:

a-c: Please specify in the figure legend what the green staining represents. Please also explain the white spots, are these fungal hyphae?

In panel a it was unclear whether the mRFP was found in all cells containing arbuscules or only some? Would it be possible to counterstain some roots with wheat germ agglutinin to test whether all cells containing arbuscules also show mRFP expression?

In Figure 6d, do the statistical differences refer to comparisons with the WT? It would really be most important to test whether the mutant (*ark1-1* EV) phenotype was significantly different from the rescued mutant (*ark1-1* ARK1-RFP). Was this done? It looks like there might be a partial rescue but not a full one. In the text, (line 354), the authors state that 'a restoration of wild-type amounts of fungal structures was observed in *ark1-1*ARK1-mRFP plants (Fig. 6d)', but this is not

the case looking at the figure, the phenotype is not fully restored.

Figure 6d-f: It was quite clear that the immunogold labelling is located on the PAM, but please provide some images where the plasma membrane or other plant membrane compartments can be seen to test how specific the localization in the PAM really is.

Supplementary Tables 1, 7, 8 and 9 require a heading and explanation of what they show. Please also specify the databases used for each of the ID names.

In Supplement Figure 2, it was surprising to see that there was a rather large difference in gene expression between cortical cells with and without arbuscules of infected roots (A vs S), but a much smaller difference between arbuscule-containing cortical cells and cortical cells of mock-infected roots (A vs M). Could the authors please speculate about possible reasons for this? I would have expected a much bigger difference between A and M.

Reviewer #2:

Remarks to the Author:

The manuscript by Roth et al describes the identification of a plant protein kinase gene ARK1 from rice via transcriptomics and proteomics of arbuscular mycorrhiza engaged roots. The phenotype of the ark1 mutant maybe different from previously characterized arbuscular mycorrhiza defective mutants that are typically characterized by arbuscule absence or structurally impaired arbuscules. In contrast, ark1 may form structurally wild-type like arbuscules but (and this is novel) does not actively support important storage organs like vesicles and spores. This is indeed an interesting novel finding that may reveal previously not detected regulatory circuits, but at the same time raises the question about the underlying mechanism that could be at least partially be answered with more detailed molecular analysis of the phenotype.

Scientific impact of the work

The authors emphasize the morphologically normal arbuscule structure of the ark1 structure. While this may be correct, it does not deliver a mechanistic explanation for the ark1 mutant phenotype. This phenotype may hypothetically be caused by a defect in arbuscule function or altered PAM composition. More insights into the mechanism would be the - in the present version sorely missing - icing on the cake. Considering the lack of vesicles and spores it appears that the arbuscules do not sufficiently feed the fungus. This could be easily tested using the same analytical techniques used in recent papers demonstrating lipid transfer from the plant to the fungus. An alternative and maybe even easier approach would be a transcriptome comparison of the fungal extraradical hyphae feeding on wild type with those feeding on the ark1 mutant. This comparison may provide insights into the missing contribution of the plant.

Nutrient transfer analyses from and towards the fungus would potentially provide interesting mechanistic insights into the role of ARK1 in arbuscule functioning. If the ARK1 phenotype affects arbuscule function, one should be able to analyse this also for arbuscules generated in the presence of nurse plants. This way more arbuscules are present but their function should still suffer from the lack of ARK1.

A second potential mechanistic aspect of ARK1 is the fact that although the arbuscules may not work properly, the plant appears not to respond with early arbuscule degeneration observed in the Medicago PT4 mutants previously described. To follow up the hypothesis that ark1 is involved in regulating the arbuscule turnover, the authors could cross ark1 mutant with Ospt11 mutant (rice ortholog of MtPT4) and determine whether the degeneration of the arbuscule determined by the pt11 mutant is altered in the double mutant.

A third potential role of ARK1 may affect the protein composition of the periarbuscular membrane.

The authors claim to have established a PAM proteome analysis pipeline. If this is true, one would expect the authors to exploit this facility to characterize ARK1 as a PAM analysis of the ark1 mutant roots. These could ideally also include the arbuscules formed in the presence of supporting wild-type nurse plants, because they are more abundant and should circumvent the obstacle of arbuscule limitation in the ark1 mutant.

Alternatively they could improve the manuscript by expand on the -omics data presented in Figure 2. The 22 proteins should be listed in the main text as a table, because that's a key finding of the paper. This could be expanded with phylogenetic trees of the candidates, to show the presence/absence of the 22 proteins in different plant lineages of interest. This could be further expanded by phenotyping of TOS17 and Medicago Tnt1 lines to strengthen the claim that their proteomics and transcriptomics effort yielded valuable information about novel players in AM symbiosis.

The authors could also include a comparison with the available peribacterial membrane proteome. At least at the level of functional annotations.

These are just a few spontaneous ideas, and not imperative requests. However, in its present form the data presented in the manuscript, although they may represent years of work, unfortunately appear rather preliminary.

Major criticisms

1) Data quality

The phenotype of the ark1 mutant has some major weaknesses. A valid quantification of spores that are located outside the root is very difficult or impossible to achieve because the hyphae will break during the removal of substrate particles and the spores will be lost in variable percentages. Therefore the spore counts are questionable. The authors should elaborate on how they ensure comparable spore retention frequencies from wild type and mutants.

Figure 5. What do the bar charts represent? Medians? In 5f, there is a big difference in the phenotype between both Ark1 mutants, that looks different in figure 5a and b, why is that? How reliable is the reproducibility of the phenotyping? 3 measuring points (independent biological replicates?) may simply not be enough considering the variation between experiments. This variation has also major implications for the complementation analysis.

Figure 6. The authors claim that "ark1 mutant is complemented by ARK1". However, the construct chosen restores only partially the wt colonization levels. The two particularly relevant parameters, arbuscule frequency and vesicle frequency are indeed approximately 30% of the wild-type levels, why? Position of the tag, promoter chosen or background mutations are some possible causes. If the promoter is incomplete, the localisation studies that use this construct potentially fail to cover the whole expression domain of ARK1. Or worse, does it mean that the phenotype is generated in combination with a second mutation in the genomes of the two underlying alleles? Or is simply the phenotyping so variable, that no conclusions can be drawn? Given this problem, they should try to improve the complementation level by reducing the tag size and increase the promoter length and try to complement both lines.

Line 251 and Figure 3. "Rice LYK1 and AM14 251 (ARK1) promoter activities are specific for arbuscule containing cells". Staining of the fungus (that can easily be done with WGA, acid fuchsin...) is mandatory to state that, and to precisely define in which cells (only in arbuscule containing cells or also in the neighbouring cells and cells containing hyphae?) and at which precise stage of the arbuscule developmental process these promoters are active. Additionally, the number plants/roots inspected and the number of detected arbuscules showing GUS expression and the number of arbuscules that does not show GUS staining must be specified for that figure.

Line 274. "but is absent from non-mycorrhizal plants species such as *Arabidopsis thaliana*" Which plants other than *Arabidopsis* were analysed? A phylogenetic tree is required to show the presence/absence of that protein among plant species of interest. These include at least rice, *Arabidopsis* and *Medicago*.

Line 286. The authors speculate about the presence of "active" kinase domain in that protein and later on discuss about the putative role of this protein in signalling at the PAM. It would be very useful to perform a kinase assay to test the functionality of this kinase domain, which would give some support to your hypothesis.

The authors claim that ARK1 is localizing to the PAM. However there are a few problems with the data that lead to this claim. In Figure 6. The antibody is labeled with gold particles and these dots should be in the intercellular space, because the hapten is fused to the C-terminus of the predicted membrane anchored kinase. However, all dots shown are in the apoplast and outside of the cytoplasm. What does the arrowheads shows? Statistic on the number of observations are missing!! In the legends, replace ARK by ARK1. In Line 346: "we observed subcellular localization of ARK1-mRFP to PAM surrounding arbuscule branches" Because they claim in line 48 and line 415-416. that "ARK1 is thus required for fungal fitness at a novel functional stage, post arbuscule development" they should perform a time-course and also localise ARK1-mRFP when the arbuscule started to degenerate.

Figure S5. The ratio should be replaced by absolute size values as in Javot et al. They should also measure the size of the cells in the *ark1* mutant. Are they similar to the ones in WT plants? Is the cell shape in the mutant inspected? The heterogeneity of the cortical cell length in their pictures suggests that not all phenotypic aspects of arbuscule degeneration may be even detectable in rice.

The authors claim that ARK1 is PAM-specific, but the analysis is limited to arbuscocytes. To maintain this claim they should also investigate other tissues and/or non-infected roots for ARK1 expression. This can easily be done with the stable transgenic lines at hand.

The TOS17 line numbers should be included to allow reproduction of the results by other labs. Also the connection between the TOS17 mutant analysis described in the PhD thesis of Caroline Gutjahr and the present work should be clarified, if appropriate by a co-authorship.

2) Manuscript structure

The authors could drastically improve their manuscript by removing the lengthy description of LYK1 mutation of which did not result in a phenotype and revive that part in a later manuscript in which they will have generated and analyzed double mutants that deal with the potential redundancy within this receptor family. At the moment this RLK consumes as many words as ARK1 but does not really provide any scientifically valuable information. Could be either completely deleted or moved to supplements.

Instead the authors should use the freed-up space to provide functional or mechanistic data on ARK1 or other aspects listed above.

3) Inaccuracies and omissions

Figure 1. Scale bar wrong? Picture of bad quality, we don't see the PAM on it. This Figure is not necessary, can move to Suppl.

Line 64. Is it really proven that the lipid transfer occurs across the PAM?

Line 73. "intractability of the fungus", not "fully" true, look at the papers from Andrea Genre for fungus localization coupled with in vivo imaging...

Line 74. What kind of “changes” are you talking about, this notion is very vague and unclear.

Line 83. “root nodulation symbiosis of legumes with nitrogen-fixing bacteria”. It is NOT only for legumes but also for other nodulating plant species that belong to other orders of the nodulating clade.

Line 85-89. “The role of membrane bound receptor-like kinases (RLK) to initiate such ‘arbuscule accommodation module’ is currently unclear. However, given the extremely dynamic, intimate nature of the arbusculated cell, the orchestration of plant-fungal signal exchange is likely to occur at the PAM.” The transition is not very clear for the reader....

Line 101. “unfolded arbuscules”, what do you mean by “unfolded”. This term is never used in the Results and Discussion part later in the text... there is no picture of “unfolded arbuscule” to exemplify this in the manuscript.

Line 103. “we provide a first insight into signalling at the PAM”, there is actually NO signalling shown in the presented manuscript. Not even a kinase assay! This is just a speculation about the “PUTATIVE” role of ARK1.

Line 107. Results section “Establishment of PAM-enriched membrane proteome workflow”, too much description of the methods, this part should rather move to the “Material and Method” section. It is boring to read and you “lose” the reader into details not always relevant for the understanding of the results generated. There is no comparison of different methods/approaches that would lead to the identification of different proteins, so no need to describe the approach in detail in the result section.

Line 261. “Molecular characterisation ...”, find another title for that section, there is no “molecular characterisation”, only speculation about the function of these proteins.

Line 418. You wrote “AM symbiosis fails to mature”. What does “mature” means, can you precise what do you mean with this term in that sentence?

Overall, the words “reviewed in” are written 6 times in the paper. Better cite the original publications and not only reviews.

Figure 4.

Panel a), add scale bar on the gene structure

Panel b) annotate “ED” in the ARK1 protein structure.

Panel e) RT-PCR performed, it will be nice to perform qRT-PCR as well. Under which conditions these RT-PCRs were done? Mock or inoculated roots, 3 or 6 wpi? RT-PCR method not described.

Figure 5. Panel c) and d). What is red? Propidium iodide staining? This information is missing in the legend and in the “Material & Method” sections. Additionally, imaging methods of WGA staining is missing in that section.

Figure S4. Show data points instead of histogram.

Figure S5. Was this performed 6 wai?

Authors are advised to follow the guidance in the instructions for authors at The Plant Cell for the correct nomenclature of DNA fusions. “::” classically stands for insertion events. What they constructed are fusions, which should be labelled by “:” Translational fusions are ideally indicated with a hyphen.

They should apply the labels consistently. In Figure 3 there is inconsistency with construct names “pARK1:GUS” or “pARK1::GUS”?

RNA stored at 80°C? Seriously?

Line 37. "fungal haustoria", is that term correct?

Line 75. "work of recent years", English correct?

Line 97. Correct "to the PAM instead of "to PAM"

Line 98. "is needed to allow the fungus to form storage vesicles", English correct?

Line 229. "(Gutjahr et al., 2015)" citation should be a number to be consistent.

Line 338. Correct "ARK1 localizes to the PAM" instead of "ARK1 localizes to PAM"

Line 375. "at present", English correct?

Line 538. The definition of the abbreviation "false-discovery rate (FDR)" should rather be placed
Line 221.

Reviewer #3:

Remarks to the Author:

This is a strong paper, representing a significant advance in an important aspect of AM symbiosis we know very little about. While we have achieved some understanding of the role of RLKs in mediating communication between the plant and its symbiont at early stages of symbiosis (ie, perception and recognition), we know next to nothing about signaling events by which plant and fungus almost certainly and continually communicate with one another to maintain a productive symbiosis at later stages of colonization. Logic implies that these signaling events likely take place at the peri-arbuscular membrane (PAM), which represents the most extensive interface between plant and fungus, however this structure is very difficult to access experimentally.

The authors present strong evidence that the RLK ARK1 is specifically expressed, localized, and associated with the peri-arbuscular membrane in rice (and in maize). The authors further demonstrate that AM colonization is perturbed in ARK1 rice mutants at a relatively late stage of symbiosis (6 weeks post-inoculation), implying a role of ARK1 in supporting the longer-term stability of the symbiosis, and in promoting fungal fitness. It is worth emphasizing that studies of this type in AM symbiosis are technically very challenging and yet the authors have (for the most part) clearly presented multiple lines of data that support their overall conclusion – that ARK1 functions to mediate signaling between plant and fungus at the PAM, and that such signaling is critical to a productive symbiosis. The role of the second (LysM)-RLK, encoded by LYK1 remains unclear and it might be argued better to remove this section and concentrate the paper entirely on ARK1, however I do feel it important to include all data presented, as it will add to an ongoing debate within the community as to the role of LysM-proteins in mediating signaling events in AM symbiosis, with a unique twist – signaling at the PAM.

General comments

While it is clear that the authors were careful to identify relevant markers to track enrichment of proteins from underrepresented membrane tissues, for example, PAM-specific proteins such as STR and PT6, what is less clear is how carefully non-membrane associated (ie, cytoplasmic) proteins were tracked as a means of estimating the inevitable 'contamination' of these within the membrane-enriched fractions. For example, in looking through the plant (and fungal) proteins in Supp Table 1, I can find some enzymes, trxn factors, uncharacterized proteins that lack canonical signal peptides and TMD, etc, that are unlikely to be membrane-bound (or directly associated with membrane proteins) and are most likely high abundance cytoplasmic proteins. It also seems as though the Blast2Go was used to assess PM enrichment against other membrane proteins, but not contaminating non-membrane proteins. While this in no way invalidates the discovery of ARK, etc, it would be informative to discuss this briefly, and useful to include in Supp Table 1 an in silico assessment of protein localization for all of the proteins listed (eg, wolf psort and TMHMM2), to help discriminate between the genuine membrane proteins and contaminating non-membrane proteins. Along this vein, in pg 8, when the authors identify for example 3640 'membrane'

proteins, has this classification been supported by in silico analyses (eg, presence of a signal peptide and TMD) or would it be more correct to classify these as 'proteins extracted from membrane-enriched fractions'? My comments apply equally to the rice root proteins (Suppl Table 7).

While perhaps not surprising, the detection of hundreds of fungal proteins from both colonized maize and rice roots is an intriguing additional data set, but is not discussed at all by the authors. While the focus of the project was to identify PAM-localized plant proteins, presumably the methods employed would have also enriched for components of the fungal membrane and its proteins?? I appreciate space limitations, but it would be worth adding a comment or two as to the nature of these fungal proteins. Do the authors expect the fungal membranes to co-purify with the PAM? and do they believe these fungal proteins to originate from the fungal membrane, or perhaps these are mostly contaminating cytoplasmic proteins? Any evidence of so-called effector proteins that might be associated with the PAM?

I am concerned about the surprisingly small number of AM-specific proteins that overlap between maize and rice (Fig 2c). Yes, some of large number of non-overlapping proteins will be attributable to species specificity (line 209; pg 10) but I do suspect much of this is due to noise...however this may be a necessary trade-off to achieve the sensitivity required for this approach to be successful.

Specific comments

Pg 6 – line 111 – yellow pigment is a bit vague. Is anything further known about this pigment? presumably mock inoculated roots do not accumulate this pigment, which raises the question of how the MI control roots were selected for harvest? Were these roots randomly chosen?

Pg 7 – line 139 – 'within the distal' I think the authors mean 'within the distal membrane'?

Pg 12 – Mol. Characterization of LYK1 - Gomez et al., 2009 presented evidence (also obtained by laser capture microdissection) of a LysM RLK (MtLYR1) that is specifically expressed in arbusculated cells – I am curious whether this is a homologue of LYK1? If so, it would be appropriate to cite this work.

Pg 13- lines 301-303, and Suppl Fig 3 – the authors demonstrate that two lines of LYK1 mutants have WT colonization levels at 6 wpi, however it seems odd that they make no comment or present data about colonization at an earlier time point (ie, 3 wpi) and/or no comment about qualitative phenotypes such as formation of arbuscules, etc. The implication in the text of the results and discussion is that the LYK1 mutants exhibited no abnormal phenotype and this may be due to redundancy with other RLKs, however it would be appropriate to clarify whether, for example, an earlier 3 wpi time point was assessed and determined to be comparable with WT.

Pg 16 – lines 354-356 – the authors complemented the ark1-1 mutant with the mRFP construct and concluded 'a restoration of wild-type amounts of fungal structures...demonstrating that the ARK1-mRFP fusion protein was functional' however looking at the data, there is statistically significant difference between the colonization profiles of WT plants and ark1-1 lines complemented with ARK1-mRFP, suggesting only partial complementation. If this is the case, the authors should modify their statement in lines 345-346 accordingly.

Pg 19 – line 449 – 100mM and 25mM KH₂PO₄? Surely it would be micromolar?

Fig 1. – last line of caption, correct 'arbuscule'

Fig 4 – I'm wondering whether this figure would be better placed in Supplemental?

Fig. 4C – part of the figure is cut off (activation loop)

We thank the editor and the reviewers for their time and effort to evaluate our work. Below, please find our point-for-point replies to their comments.

Reviewer #1 acknowledges the identification of the receptor-like kinase ARK1 with a new function in AM symbiosis, and considers this finding as important, paving the way for future studies of signalling mechanisms underpinning the maintenance of the symbiosis.

General Comments:

1. Somewhere in the results (supplement) the authors need to show the details of the protein identification, i.e. a list of proteins with the number of peptides that each protein was identified with, the peptide sequences, the number of mis-cleavages and allowed modifications and the mass accuracy.

We have produced a summary table (Dataset 1) containing the requested information, including identification probability, exclusive unique peptide counts, exclusive unique spectrum counts, percentage of total spectra, percentage sequence coverage, peptide sequence, best peptide identification probability, etc. In addition, we have submitted the MS proteomics data to ProteomeXchange via the PRIDE database under the ID of PXD002575 for maize and PXD010313 for rice. The following text has been added to the Materials and Methods “The mass spectrometry proteomics data have been deposited to the ProteomeXchange Consortium (<http://proteomecentral.proteomexchange.org>) via the PRIDE partner repository with the dataset identifier PXD002575 for maize and PXD010313 for rice.” (page 23, lines 531-534).

2. A lot more proteins were identified to be enhanced in mycorrhizal roots compared to mock inoculated roots in the whole root approach compared to micro-dissected cells. Only 43 transcripts were found to be differentially accumulated in arbuscule-containing cells (line 222), while in the proteomic approach, 1180 proteins were stated to be exclusively found in colonized roots (line 199). Could you discuss the likely function/location of the remaining differentially accumulated proteins?

The preservation of the RNA is compromised during the lengthy manipulations of the tissue for laser capture microdissection (LCM), explaining the quantitative difference between the number of differentially expressed (d.e.) proteins relative to d.e. ‘LCM transcripts’. In addition, the proteome analysis was performed on whole root tissue and would therefore contain proteins that differentially accumulated at other stages of the interaction than exclusively in arbusculated cells. To this end, the LCM transcriptome enabled the discrimination of genes expressed in arbusculated cortex cells.

3. A lot more rice proteins were identified than maize proteins. Could you discuss the reasons for this?

We agree with the reviewer that this is initially surprising, but propose the better quality of the rice genome annotation as an explanation.

4. Overall there was a surprisingly small overlap between proteins and transcripts detected in mycorrhizal roots/arbuscule-containing cells. The authors did not really speculate on this but something should be included in the discussion; i.e. what are all the other almost 500 / 900 proteins that were detected in the proteome analysis of microsomal enriched mycorrhized roots compared to the arbuscule containing cells?

As mentioned in our reply to comment 2 the quality of the proteome dataset would be considered superior to that of the LCM transcriptome. The following comment has been added to the results section “A considerable overlap was observed between the differentially expressed genes identified

and our published analysis of whole roots, indicating the robustness of the LCM transcriptome dataset” (pages 9-10, line 197-199).

Specific comments:

5. Line 180: Previous reports are mentioned here but not referenced. Please include a reference to studies that this one is compared to.

The references have been added as per the reviewer’s suggestion (page 8, line 150).

6. Line 304: Please check the end of this sentence (delete ‘at’?) and define ‘wpi’ at the first mention: ‘at six wpi both ark1-1 and ark1-2 mutants displayed significantly reduced levels of all fungal structures, whereas at three weeks post inoculation (wpi) both ark1 mutant lines revealed equivalent levels of most fungal structures relative to the wild-type at’

The text has been amended according to reviewer’s suggestion (Page 13, Line 276-276).

7. Line 208: ‘The relatively large number of 524 and 1011 proteins not shared between maize and rice (Fig. 2c) could represent species-specific proteins in addition to technical noise.’ What do the author refer to with ‘technical noise’? Have other proteomics studies found large numbers of species-specific proteins? Could the reason include the naming of proteins in different databases (i.e. proteins with the same functions could have different names and not be matched).

As “technical noise” we intended to point towards the problem that finding the correct protein match between two species can sometimes be compromised, arising from multiple isoforms, hypothetical protein, etc. We rephrased the sentence to “technical noise arising from difficulties finding the correct protein match between two species due to e.g. the existence of multiple isoforms to avoid confusion (page 9, line 179-180).

8. Line 242: ‘LYK1-mRFP fusion protein was not detectable in 8 independent colonized lines (data not shown).’ What could be the reason?

We propose low overall protein expression levels below detection levels as an explanation.

1-9. Line 333: When saying ‘by growing R. irregularis inoculated mutant and wild-type plants in the same compartment’, does this mean that the fungus can form a symbiosis with both WT and mutant plants at the same time by growing hyphae concurrently into each root?

Indeed, the fungus has the ability to colonise multiple plants at the same time, thereby producing a so called common mycorrhizal network. For more information, please see Walder & van der Heijden, Nature Plants 2015.

10. Line 363: OsPT11-eGFP was used as a control for RFP antibody specificity, why was this construct chosen?

As the anti-RFP anti-bodies do not cross-react with GFP, the PT11-eGFP expressing provided a negative control to assess anti-body specificity.

11. Line 444: ‘Wild-type nurse plant inoculation was performed in 4.5cm petri dishes to achieve rapid colonization within 20 days post inoculation’ Were maize or rice plants indeed grown in 4.5 cm petri dishes for this experiment? It was hard to visualize them fitting into those plates without restricting the root growth.

Only rice seedlings, not maize, were grown using this nurse plant system with 4.5mm petri-dishes to enable rapid colonisation of 20 days old seedlings for imaging purposes (TEM and live cell imaging).

Physiological nurse plant experiments were grown in pots. The text was rephrased for clarification "For TEM and live cell imaging, to ensure rapid colonization within 20days post inoculation (dpi), pre-germinated rice seedlings were grown in the presence of a nurse plant in 4.5cm petri dishes." (Page 18, line 408-410.)

12. Line 489: 'The used concentrations were 8µg'. 8 µg is not a concentration but an amount.

We have corrected the sentence to: "The used concentrations were 8µg/µl for PMPFs." (page 20, line 456).

13. Line 500: Please detail the conditions used for HPLC and coupled mass spectrometry in the Orbitrap mass analyzer.

The requested information has been added to the Materials & Methods section (page 21, lines 489-502).

14. Line 547: this should probably read '-80 degrees'.

The text has been corrected (page 25, line 562).

15. Figure 1 was very helpful!

*We agree with the reviewer that the figure is useful but have moved it to the Supplementary Information, now appearing as **Supplementary Fig 1** according to the suggestion of reviewer #3.*

16. Figure 2: I did not completely understand this figure: In panels a and b, what are the numbers in the circles, and why are there two rows for each, e.g. Plant proteins: 776 – 15.6%, und underneath: 91 – 1.8%? Please explain this in the legend.

The legend contains the requested explanation "underlined numbers indicate the quantified proteins"; i.e. 776 refers to the number of proteins detected and 91 to the number of quantified proteins.

17. Figure 3: It appears from the white arrow indicating arbuscules that the GUS staining is not specific for cells containing arbuscules. Is this correct, or are not all arbuscule-containing cells marked with an arrow? The roots need to be counterstained with ink (or similar) to indicate GUS staining at the same time as arbuscules.

*According to reviewer's suggestion Figure 3 (now Figure 2) has been amended to include WGA staining and numbered arbuscules, confirming that GUS staining only occurs in arbuscule containing cells of the transgenic line expressing PromPT11:GUS. To accommodate the second reviewer's suggestion, the PromLYK1:GUS data have been moved to the Supplementary Information and now correspond to **Supplementary Fig. 4**.*

18. Figure 5f: I assume that the significant differences are found between the first three bars and the last two bars, is this correct? It might be easier to indicate this with letters, e.g. the label the first three bars with 'a' and the last two with 'b'.

*We agree with the reviewer and have made the following changes; we now plot the quantity of root length colonisation uniformly as a function of genotype and performed separate Kruskal-Wallis tests, using the post hoc Dunn's tests and Benjamini-Hochberg for multi-comparison adjustment. Former Fig. 5f appears as **Fig. 3f** in the revised version of the manuscript.*

19. Figure 6: a-c: Please specify in the figure legend what the green staining represents. Please also explain the white spots, are these fungal hyphae?

The green colour corresponds to auto fluorescing plant cell walls, and the white spots to frequently found auto fluorescing 'plant bodies' of unknown identity or function. The figure legend of Fig. 6 (now Fig. 4) has been amended for clarification and now reads as follows: 'Auto fluorescence from plant cell walls are false colored in green, while auto fluorescent plant bodies present in both channels are shown in white.'

20. In panel a it was unclear whether the mRFP was found in all cells containing arbuscules or only some? Would it be possible to counterstain some roots with wheat germ agglutinin to test whether all cells containing arbuscules also show mRFP expression?

Live cell imaging does unfortunately not permit simultaneously staining with WGA as the latter is a destructive technique requiring tissue fixation. However, live-cell imaging of mature arbuscules shows weak levels of autofluorescence allowing detection of arbuscules (see Fig 4c mycorrhizal non-transformed WT control roots). We did not observe cells containing arbuscules that lacked mRFP expression.

21. In Figure 6d, do the statistical differences refer to comparisons with the WT? It would really be most important to test whether the mutant (ark1-1 EV) phenotype was significantly different from the rescued mutant (ark1-1 ARK1-RFP). Was this done? It looks like there might be a partial rescue but not a full one. In the text, (line 354), the authors state that 'a restoration of wild-type amounts of fungal structures was observed in ark1-1ARK1-mRFP plants (Fig. 6d)', but this is not the case looking at the figure, the phenotype is not fully restored.

We apologise for this oversight. The text has been amended accordingly to "significantly higher levels of colonization was observed in ark1-1^{ARK1-mRFP} compared to ark1-1^{ev} control lines (Fig. 4e), demonstrating at least partial functionality of the ARK1-mRFP fusion protein". (Page 15, Line 323-325)

22. Figure 6d-f: It was quite clear that the immunogold labelling is located on the PAM, but please provide some images where the plasma membrane or other plant membrane compartments can be seen to test how specific the localization in the PAM really is.

To document the specificity of the antibody relative to other endomembrane compartments, we have included arrows to point out ER and tonoplast membranes not showing any label (Figure 4f,g), and have accordingly explained this in the figure legend.

23. Supplementary Tables 1, 7, 8 and 9 require a heading and explanation of what they show. Please also specify the databases used for each of the ID names.

Headings and explanations have been added to the indicated supplementary tables.

24. In Supplement Figure 2, it was surprising to see that there was a rather large difference in gene expression between cortical cells with and without arbuscules of infected roots (A vs S), but a much smaller difference between arbuscule-containing cortical cells and cortical cells of mock-infected roots (A vs M). Could the authors please speculate about possible reasons for this? I would have expected a much bigger difference between A and M.

There was indeed an error in the analysis, and the subsequent preparation of the figure, for which we apologise. This has been corrected. Differences between A vs S and A vs M may reflect the level of variability observed within the S and M samples, itself likely the result of the technical difficulties of

LCM. A consequence of such differences may be transcripts called in one contrast that are close to, but not above, the threshold in the second contrast, giving a slightly misleading impression of how similar or otherwise samples may be. We have adjusted our analysis, using calling methods and thresholds that, on inspection, appear to be less sensitive to such effects. Notably, we recover substantial overlap between A vs S and A vs M contrasts. We emphasize that the primary goal of this analysis was to provide a further filter of candidates identified in the proteomic study, prior to selection of candidates for functional validation. We have provided a more extensive description of the transcriptome analysis in the methods and included additional supplemental tables of these results (Page 24-25, Line 570-587, and associated Supplemental Tables 10 and 11).

We are pleased that **Reviewer #2** recognizes the novelty of the phenotype associated with the mutation of *ARK1* and thereby the potential for gaining new insights into regulatory circuits operating during AM symbiosis establishment, post arbuscule development. His/her preference for more mechanistic data is certainly legitimate. However we feel that such data is beyond the scope of the current study, an opinion which we consider to be shared by the other reviewers from who the opinion of reviewer #2 diverges.

General Comments:

1. The manuscript by Roth et al describes the identification of a plant protein kinase gene *ARK1* from rice via transcriptomics and proteomics of arbuscular mycorrhiza engaged roots.

We would like to emphasise that maize proteomics played a major role in the identification of ARK1. The dataset therefore includes maize and rice proteomes of mycorrhizal and non-mycorrhizal roots, and in addition the transcriptome of arbusculated rice cells, obtained from laser captured material.

2. The phenotype of the *ark1* mutant maybe different from previously characterized arbuscular mycorrhiza defective mutants that are typically characterized by arbuscule absence or structurally impaired arbuscules. In contrast, *ark1* may form structurally wild-type like arbuscules but (and this is novel) does not actively support important storage organs like vesicles and spores. This is indeed an interesting novel finding that may reveal previously not detected regulatory circuits, but at the same time raises the question about the underlying mechanism that could be at least partially be answered with more detailed molecular analysis of the phenotype. The authors emphasize the morphologically normal arbuscule structure of the *ark1* structure. While this may be correct, it does not deliver a mechanistic explanation for the *ark1* mutant phenotype.

While we agree with the reviewer that it will indeed be exciting to unravel the mechanism of ARK1 functioning, such work would be beyond the scope of this manuscript, causing a delay in publishing of the new insights presented here, which we consider not to be desirable.

3. This phenotype may hypothetically be caused by a defect in arbuscule function or altered PAM composition.

The reviewer uses the rather vague term of 'arbuscule function'. It is not entirely clear what s/he is referring to as we report on a plant not a fungal mutant; fungal mutants are unfortunately not existing in the absence of genetic protocols for the fungal system.

An altered PAM composition as a result of mutation of ARK1 is an attractive hypothesis that however cannot easily be addressed. Please also see answer to comment 6 of this reviewer.

4. More insights into the mechanism would be the - in the present version sorely missing - icing on the cake. Considering the lack of vesicles and spores it appears that the arbuscules do not sufficiently feed the fungus. This could be easily tested using the same analytical techniques used in recent papers demonstrating lipid transfer from the plant to the fungus. An alternative and maybe even easier approach would be a transcriptome comparison of the fungal extraradical hyphae feeding on wild type with those feeding on the ark1 mutant. This comparison may provide insights into the missing contribution of the plant. Nutrient transfer analyses from and towards the fungus would potentially provide interesting mechanistic insights into the role of ARK1 in arbuscule functioning. If the ARK1 phenotype affects arbuscule function, one should be able to analyse this also for arbuscules generated in the presence of nurse plants. This way more arbuscules are present but their function should still suffer from the lack of ARK1.

Please compare answer to comment 2 of this reviewer. In addition, we would like to remind the reviewer that it has been well documented in the literature that dysfunctional lipid feeding of the fungus consistently resulted in stunted arbuscules (Bravo et al. 2017; Jiang et al., 2017; Keymer et al., 2017; Luginbuehl et al., 2017). In contrast, in rice ark1 mutants there is no evidence for abnormal arbuscule development. We would like to reiterate that the scope of this manuscript is the introduction of a fundamentally new concept of a plant regulatory step in AM symbiosis post arbuscule development that - in our view - would merit the swift communication to the field.

5. A second potential mechanistic aspect of ARK1 is the fact that although the arbuscules may not work properly, the plant appears not to respond with early arbuscule degeneration observed in the Medicago PT4 mutants previously described. To follow up the hypothesis that ark1 is involved in regulating the arbuscule turnover, the authors could cross ark1 mutant with Ospt11 mutant (rice ortholog of MtPT4) and determine whether the degeneration of the arbuscule determined by the pt11 mutant is altered in the double mutant.

We appreciate the efforts the reviewer undertakes in proposing mechanistic scenarios, but wish to clarify that we did not hypothesise that ARK1 is involved in arbuscule turnover. We assume that when speaking about “the arbuscules may not work properly” s/he refers to a defect in symbiotic phosphate uptake on the plant side as fungal mutants are not existing (see our reply to comment 3 of this reviewer). Importantly, if phosphate uptake at the PAM is perturbed, arbuscules fail to fully expand and appear crippled (Javot et al., 2007; Yang et al., 2012). However, in ark1 mutants arbuscule morphology, arbuscule size, and arbuscule turnover rates are equivalent relative to WT. In summary, we consider the suggested experiments as interesting but not directly relevant for this manuscript.

6. A third potential role of ARK1 may affect the protein composition of the periarbuscular membrane. The authors claim to have established a PAM proteome analysis pipeline. If this is true, one would expect the authors to exploit this facility to characterize ARK1 is a PAM analysis of the ark1 mutant roots. These could ideally also include the arbuscules formed in the presence of supporting wild-type nurse plants, because they are more abundant and should circumvent the obstacle of arbuscule limitation in the ark1 mutant.

We fully agree with the reviewer but feel a need to clarify that the proteomics work was carried out at the Functional Genomics Centre in Switzerland while first and last author, MC and UP, were Swiss employees at the University of Lausanne. Since leaving Switzerland, funding for additional proteomics efforts has unfortunately not been available.

7. Alternatively they could improve the manuscript by expand on the -omics data presented in Figure 2. The 22 proteins should be listed in the main text as a table, because that's a key finding of the paper.

We thank the reviewer for this suggestion. Table 2 of the revised manuscript now displays the genes/proteins consistently recovered as differentially expressed from maize proteomics, rice proteomics and rice LCM approaches.

8. This could be expanded with phylogenetic trees of the candidates, to show the presence/absence of the 22 proteins in different plant lineages of interest. This could be further expanded by phenotyping of TOS17 and Medicago Tnt1 lines to strengthen the claim that their proteomics and transcriptomics effort yielded valuable information about novel players in AM symbiosis. The authors could also include a comparison with the available peribacterial membrane proteome. At least at the level of functional annotations.

We concur with the overall strategy suggested by the reviewer, namely protein identification followed by phylogenetic analyses and reverse genetics. As the objective of this study was to find PAM-associated putative signalling components (e.g. receptor-like kinases), we focussed on LYK1 and ARK1, and indeed applied exactly this strategy. To extend this approach to all identified proteins as suggested by the reviewer would in our view however, address an entirely different question.

9. These are just a few spontaneous ideas, and not imperative requests. However, in its present form the data presented in the manuscript, although they may represent years of work, unfortunately appear rather preliminary.

We disagree with the reviewer's judgement and for comparison would like to draw his/her attention to the publication of another receptor-like kinase, SYMRK, known for being essential for AM symbiosis establishment, in contrast to ARK1 however operating at early stages of the interaction. The original work by Stracke et al. (Nature 417, 459) reported the phenotype, gene cloning and expression patterns. Even in the absence of mechanistic information, this article has provided invaluable 'food for thought' although still today, 16 years later, we do not understand SYMRK function.

Specific comments:

10. The phenotype of the ark1 mutant has some major weaknesses. A valid quantification of spores that are located outside the root is very difficult or impossible to achieve because the hyphae will break during the removal of substrate particles and the spores will be lost in variable percentages. Therefore the spore counts are questionable. The authors should elaborate on how they ensure comparable spore retention frequencies from wild type and mutants.

*We agree with the reviewer that control over uniform spore retention is difficult to achieve and have adjusted the text accordingly. **Page 3, line 42; Page 14, line 283; Page 17, line 369.***

11. Figure 5. What do the bar charts represent? Medians? In 5f, there is a big difference in the phenotype between both Ark1 mutants, that looks different in figure 5a and b, why is that? How reliable is the reproducibility of the phenotyping? 3 measuring points (independent biological replicates?) may simply not be enough considering the variation between experiments. This variation has also major implications for the complementation analysis.

The development of AM symbioses is a highly dynamic process, often resulting in variable progression of fungal colonisation between individual plants and experiments. Furthermore, also the

ark1 phenotype is dynamic and only becomes apparent after the first arbuscules had been produced, coinciding with rapid vesicle (and spore) formation in the wild type, which is significantly reduced in the mutant and is accompanied by an overall decrease in root colonisation levels. The correct scoring of the *ark1* phenotype requires experience.

Regarding the rather surprising request of the reviewer to comment on the reliability of our phenotyping data, in addition to presenting phenotypic data from multiple independent experiments here, we have consistently observed this phenotype in more than ten independent experiments over the past four years. Moreover, as mentioned in our manuscript (**page 17, line 387-390**), mutation in the homologous *Medicago truncatula* *Kin3* gene led to a comparable phenotype with no difference in colonisation levels at 3wpi but a significant reduction at 5wpi (Bravo et al., 2016).

The data presented refer to independent experiments with three, four and ten biological replicates. We have rearranged **Fig. 4a and 4b** to emphasise the comparison of genotypes within experiments rather than across specific time points and experiments.

12. Figure 6. The authors claim that “*ark1* mutant is complemented by ARK1”. However, the construct chosen restores only partially the wt colonization levels. The two particularly relevant parameters, arbuscule frequency and vesicle frequency are indeed approximately 30% of the wild-type levels, why? Position of the tag, promoter chosen or background mutations are some possible causes. If the promoter is incomplete, the localisation studies that use this construct potentially fail to cover the whole expression domain of ARK1.

Please compare reply to comment 21 of reviewer 1. The text has been amended accordingly to “significantly higher levels of colonization was observed in ark1-1^{ARK1-mRFP} compared to ark1-1^{ev} control lines (Fig. 4e), demonstrating at least partial functionality of the ARK1-mRFP fusion protein”. (Page 15, Line 323-325)

13. Or worse, does it mean that the phenotype is generated in combination with a second mutation in the genomes of the two underlying alleles? Or is simply the phenotyping so variable, that no conclusions can be drawn? Given this problem, they should try to improve the complementation level by reducing the tag size and increase the promoter length and try to complement both lines.

*It is disappointing that the reviewer prefers to ignore the equivalent phenotype of the two independent mutant alleles, and the confirmatory phenotype of the *M. truncatula* *kin3* mutant. While the possibility of a second mutation might be a valid criticism if a single allele was presented, the rationale for characterizing multiple alleles is precisely that the probability that the two independently generated *ark1* lines both carry independent secondary mutations presenting a similar AM phenotype is incredibly low.*

14. Line 251 and Figure 3. “Rice LYK1 and AM14 251 (ARK1) promoter activities are specific for arbuscule containing cells”. Staining of the fungus (that can easily be done with WGA, acid fuchsin...) is mandatory to state that, and to precisely define in which cells (only in arbuscule containing cells or also in the neighbouring cells and cells containing hyphae?) and at which precise stage of the arbuscule developmental process these promoters are active. Additionally, the number plants/roots inspected and the number of detected arbuscules showing GUS expression and the number of arbuscules that does not show GUS staining must be specified for that figure.

Fig. 2 has been amended, now including WGA staining (Fig. 2b), depicting intraradical hyphae and arbuscules. Arbusculated cells have been numbered to facilitate the comparison between WGA and GUS staining patterns, which indeed revealed that GUS staining is observed in all of the arbusculated

cells but not in non-arbusculated cells. To date, we did not observe arbusculated cells that lacked GUS expression. A minimum of three roots were analysed from five replicate plants.

15. Line 274. “but is absent from non-mycorrhizal plants species such as Arabidopsis thaliana” Which plants other than Arabidopsis were analysed? A phylogenetic tree is required to show the presence/absence of that protein among plant species of interest. These include at least rice, Arabidopsis and Medicago.

It has escaped the reviewer’s attention that the manuscript included a reference to the homologous MtKin3 from the recently published phylogenomics article by Bravo et al. (2016, Fig. 1f), which provides much more than the reviewer’s request for rice, Medicago and Arabidopsis as it included 38 host and 11 non-host genomes. It can therefore be confidently concluded that ARK1 is associated with mycorrhizal host but absent from non-host plant genomes.

16. Line 286. The authors speculate about the presence of “active” kinase domain in that protein and later on discuss about the putative role of this protein in signalling at the PAM. It would be very useful to perform a kinase assay to test the functionality of this kinase domain, which would give some support to your hypothesis.

We agree that in future it will be important to determine whether ARK1 would be directly or indirectly (e.g. via interaction with other signalling components) involved in signalling. However, confirmation of activity in a kinase assay would - in our view - not provide a significant advance in understanding at this stage. For comparison we would like to again refer to the publishing of SYMRK (Stracke et al. Nature 417, 459), which similarly described the intracellular kinase domain on the basis of sequence homology with other RLKs, yet without providing experimental support.

17. The authors claim that ARK1 is localizing to the PAM. However there are a few problems with the data that lead to this claim. In Figure 6. The antibody is labeled with gold particles and these dots should be in the intercellular space, because the hapten is fused to the C-terminus of the predicted membrane anchored kinase. However, all dots shown are in the apoplast and outside of the cytoplasm. What does the arrowheads shows? Statistic on the number of observations are missing!!

We would like to remind the reviewer that membranes that are obliquely sectioned will not necessarily show the gold particles of C-terminally tagged fusion proteins as cytosolic. We invite the reviewer to consult with other IGL documentations of plant membrane proteins (e.g. Wang et al., Plant Cell 2009; Viotti et al., Plant Cell 2010; 2013), reproducing a comparable distribution pattern of gold particles to ours in the area of the respective membranes.

The request for statistics is rather surprising as the technical challenges of producing high quality IGL data on ephemeral structures such as the PAM are rather well known. This is also reflected by the absence of literature reports on IGL results on PAM-intrinsic proteins. To our knowledge we are thus presenting the first IGL data of a PAM protein.

18. In the legends, replace ARK by ARK1. In Line 346: “we observed subcellular localization of ARK1-mRFP to PAM surrounding arbuscule branches” Because they claim in line 48 and line 415-416. that “ARK1 is thus required for fungal fitness at a novel functional stage, post arbuscule development” they should perform a time-course and also localise ARK1-mRFP when the arbuscule started to degenerate.

We have added a confocal image that shows ARK1-mRFP localization at collapsing arbuscules as a new Fig. 4b.

19. Figure S5. The ratio should be replaced by absolute size values as in Javot et al. They should also measure the size of the cells in the ark1 mutant. Are they similar to the ones in WT plants? Is the cell shape in the mutant inspected? The heterogeneity of the cortical cell length in their pictures suggests that not all phenotypic aspects of arbuscule degeneration may be even detectable in rice.

The ratios reported are calculated based on both the area of the arbuscules and the cortical cells containing them, therefore the size of the cells in the ark1 mutant was indeed measured. Although cortical cells in rice have diverse sizes and shapes, are all competent for hosting arbuscules. Neither arbusculated cell size nor arbuscule size differed between wild-type and mutant plants and therefore it was decided to report the ratios.

20. The authors claim that ARK1 is PAM-specific, but the analysis is limited to arbuscocytes. To maintain this claim they should also investigate other tissues and/or non-infected roots for ARK1 expression. This can easily be done with the stable transgenic lines at hand.

GUS reporter lines showed promoter activity only in arbuscule containing cells whereas non-arbusculated, neighboring cells remained unstained (Fig 2a,c,d). Similarly in ARK1-mRFP lines live cell imaging showed no signal in non-arbusculated cells (Fig. 4a-c) and RFP signal was also not detected in mock inoculated (MI) ARK1-mRFP control plants (Fig. 4d).

21. The TOS17 line numbers should be included to allow reproduction of the results by other labs.

The Tos17 line numbers have been included in the results section of the revised manuscript (page 12, lines 265-268).

22. Also the connection between the TOS17 mutant analysis described in the PhD thesis of Caroline Gutjahr and the present work should be clarified, if appropriate by a co-authorship.

Please find a copy of the section of Dr Gutjahr's PhD thesis (2009) that refers to the phenotypic characterisation of the ark1 (= AM14) alleles included below. Dr Gutjahr did parallel reverse genetics efforts on genes she had described as specifically transcribed during AM symbiosis (Gutjahr et al., 2008). She reports that she failed to observe a phenotype. The raw data were not included within the thesis, and are hence not available to me. Importantly, the ark1 phenotype can easily be missed when the co-cultivation period is too short, which is likely what happened in Dr Gutjahr's experiment. Back in 2008, we believed in genetic redundancy as an explanation and did not pursue any further reverse genetics. However, when we years later found that ARK1 was reproducibly detected in maize and rice proteomics and in the LCM transcriptomics, we were encouraged to revisit the insertion mutants. New seeds had to be ordered as an increase in seed stock had not been considered in 2008. Therefore Dr Gutjahr did not contribute data, material or 'knowledge' to this manuscript, which makes the justification for co-authorship difficult.

I would like to add that I find this reviewer comment disturbing as we consider it inappropriate to discuss co-authorship in this format.

Table 3.2. Available insertion lines with insertions in rice AM marker genes assessed for AM phenotypes. DJ, cv. Dongjin; N, cv. Nipponbare; T, cv. Tainung

Gene ID	TIGR ID	Insertion line	Insertion collection	Type of insertion	Location of insertion	Phenotype
OsAM3	LOC_Os01g57400	NF6830 (N)	NIAS, Japan	Tos17	-423 from ATG	cDNA fully expressed
		CU325387 (N)	CIRAD, France	Tos17	-152 from ATG	no insertion found
OsAM14	LOC_Os11g26140	3D-50488 (DJ)	POSTECH, Korea	T-DNA	-310 from ATG	WT-like
		NF1782 (N)	NIAS, Japan	Tos17	exon 4 of 8	WT-like
		NF4582 (N)	NIAS, Japan	Tos17	exon 6 of 8	WT-like
OsAM18	LOC_Os03g40080	NE4520 (N)	NIAS, Japan	Tos17	exon 5 of 8	WT-like
		ER893253 (N)	CIRAD, France	Tos17	exon 1 of 8	no insertion found
OsAM20	LOC_Os04g21160	1B-12003 (DJ)	POSTECH, Korea	T-DNA	intron 4 of 9	WT-like
		CL521474-6 (N)	CIRAD, France	Tos17	intron 4 of 9	WT-like
OsAM42	LOC_Os03g38600	CZ553405 (T)	TRIM, Taiwan	T-DNA	intron 1 of 11	WT-like

As shown in Table 3.2., all insertion lines had a phenotype that resembled the wild type. For *OsAM14* and *OsAM18*, that carry insertions in exons this is likely due to genetic redundancy with homologous genes. *OsAM14* encodes a serine-threonine kinase and *OsAM18* a GRAS transcription factor. Both belong to large gene families with approximately 64 and 43 members in rice, respectively. The insertions into *OsAM20* and *OsAM42* are positioned in

23. The authors could drastically improve their manuscript by removing the lengthy description of LYK1 mutation of which did not result in a phenotype and revive that part in a later manuscript in which they will have generated and analyzed double mutants that deal with the potential redundancy within this receptor family. At the moment this RLK consumes as many words as ARK1 but does not really provide any scientifically valuable information. Could be either completely deleted or moved to supplements.

We concur with the comment of reviewer #3 that the LYK1 data should be mentioned because of the interest of the field in LysM receptor kinases. Nevertheless, we do agree with reviewer2 that they are less important than the ARK1 results and have therefore moved them to the Supplementary Information (Supplementary Figure 4 and 5).

24. Instead the authors should use the freed-up space to provide functional or mechanistic data on ARK1 or other aspects listed above.

See replies above, especially to this reviewer's comments 2 and 4.

25. Figure 1. Scale bar wrong?

The scale was indeed wrong and has been corrected (new Supplementary Fig.1).

26. Picture of bad quality, we don't see the PAM on it.

The reviewer is under the wrong impression that standard light microscopy of trypan blue stained roots would permit the detection of the PAM. The image is meant to illustrate a fungal arbuscule for the broader readership.

27. This Figure is not necessary, can move to Suppl.

*We have placed the figure in the Supplementary Information as the new **Supplementary Fig.1**.*

28. Line 64. Is it really proven that the lipid transfer occurs across the PAM?

We have rephrased the sentence to account for the uncertainty (page 4, line 62).

29. Line 73. "intractability of the fungus", not "fully" true, look at the papers from Andrea Genre for fungus localization coupled with in vivo imaging...

We have rephrased the sentence to specify the 'genetic intractability' of the fungus (page 4, line 71).

30. Line 74. What kind of "changes" are you talking about, this notion is very vague and unclear.

We have rephrased the sentence to specify the 'molecular changes' that occur (page 4, line 72).

31. Line 83. "root nodulation symbiosis of legumes with nitrogen-fixing bacteria". It is NOT only for legumes but also for other nodulating plant species that belong to other orders of the nodulating clade.

We have accordingly rephrased the sentence to "nodulation symbiosis with nitrogen-fixing bacteria" (page 5, line 81).

32. Line 85-89. "The role of membrane bound receptor-like kinases (RLK) to initiate such 'arbuscule accommodation module' is currently unclear. However, given the extremely dynamic, intimate nature of the arbusculated cell, the orchestration of plant-fungal signal exchange is likely to occur at the PAM." The transition is not very clear for the reader....

We have rephrased the section of the text to facilitate comprehension (page 5, line 80-85).

33. Line 101. "unfolded arbuscules", what do you mean by "unfolded". This term is never used in the Results and Discussion part later in the text... there is no picture of "unfolded arbuscule" to exemplify this in the manuscript.

We have replaced unfolded with fully developed (page 5, line 92).

34. Line 103. "we provide a first insight into signalling at the PAM", there is actually NO signalling shown in the presented manuscript. Not even a kinase assay! This is just a speculation about the "PUTATIVE" role of ARK1.

We have changed the statement to "Together, our results suggest a role for ARK1 in signalling at the PAM and propose ARK1 function to be essential for the support of fungal fitness." (page 5, line 94-96).

35. Line 107. Results section "Establishment of PAM-enriched membrane proteome workflow", too much description of the methods, this part should rather move to the "Material and Method" section. It is boring to read and you "lose" the reader into details not always relevant for the understanding of the results generated. There is no comparison of different methods/approaches

that would lead to the identification of different proteins, so no need to describe the approach in detail in the result section.

The section has been moved to the materials and methods according to reviewer's suggestion (page 20, line 459-479).

36. Line 261. "Molecular characterisation ...", find another title for that section, there is no "molecular characterisation", only speculation about the function of these proteins.

We have rephrased the text to "Gene and protein structure of LYK1 and ARK1" (page 11, line 231).

37. Line 418. You wrote "AM symbiosis fails to mature". What does "mature" means, can you precise what do you mean with this term in that sentence?

We have rephrased the text to "In rice ark1 mutants, the low vesicle number indicates that the symbiosis fails to mature, (page 17, line 384-385).

38. Overall, the words "reviewed in" are written 6 times in the paper. Better cite the original publications and not only reviews.

The original research articles have been included where appropriate.

39. Panel a), add scale bar on the gene structure

Panel b) annotate "ED" in the ARK1 protein structure.

Panel e) RT-PCR performed, it will be nice to perform qRT-PCR as well. Under which conditions these RT-PCRs were done? Mock or inoculated roots, 3 or 6 wpi? RT-PCR method not described.

The scale bar and ED annotation has been added to the gene structure (new Supplementary Figures 5 and 6), and figure legend as well as methods section updated to provide details as suggested (page 25, line 592-593).

40. Figure 5. Panel c) and d). What is red? Propidium iodide staining? This information is missing in the legend and in the "Material & Method" sections. Additionally, imaging methods of WGA staining is missing in that section.

The figure legend of (new) Fig. 3 and the Materials and Method section have been amended as per reviewer's comments. (page 23, line 543-544).

41. Figure S4. Show data points instead of histogram.

Supplementary Fig. 4 now corresponds to Supplementary Fig.7 and has been amended to show data points as suggested

42. Figure S5. Was this performed 6 wai?

The figure legend (of new Supplementary Fig. 8) has been amended and now contains the requested information.

43. Authors are advised to follow the guidance in the instructions for authors at The Plant Cell for the correct nomenclature of DNA fusions. "::<" classically stands for insertion events. What they constructed are fusions, which should be labelled by ":" Translational fusions are ideally indicated with a hyphen.

They should apply the labels consistently. In Figure 3 there is inconsistency with construct names "pARK1:GUS" or "pARK1::GUS"?

*The manuscript text and all figures and figure legends have been verified and corrected where necessary. **Page 11, line 224***

44. RNA stored at 80°C? Seriously?

*The text has been corrected (**page 24, line 562**).*

45. Line 37. “fungal haustoria”, is that term correct?

Page 3, Line 37 changed to ‘feeding structures’

Line 75. “work of recent years”, English correct?

Page 4-5, Line 3-74 changed to ‘recent studies have’.

Line 97. Correct “to the PAM instead of “to PAM”

Page 5, Line 89 The sentence has been corrected.

Line 98. “is needed to allow the fungus to form storage vesicles”, English correct?

The English is correct.

Line 229. “(Gutjahr et al., 2015)” citation should be a number to be consistent.

The citation has been reformatted

Line 338. Correct “ARK1 localizes to the PAM” instead of “ARK1 localizes to PAM”

*The sentence has been corrected (**page 14, line 307**).*

Line 375. “at present”, English correct?

(page 15, line 342) “at present” has been replaced with ‘is currently’.

Line 538. The definition of the abbreviation “false-discovery rate (FDR)” should rather be placed Line 221.

*False-discovery rate (FDR) is now defined upon first mentioning on (**page 9, line 192-193**).*

Reviewer #3 considers our manuscript as ‘a strong paper, representing a significant advance in an important aspect of AM symbiosis we know very little about’. S/he emphasises that ‘we know next to nothing about signaling events by which plant and fungus almost certainly and continually communicate with one another to maintain a productive symbiosis at later stages of colonization.’ S/he acknowledges the experimental quality of our data that support our overall conclusion.

General comments

1. The role of the second (LysM)-RLK, encoded by LYK1 remains unclear and it might be argued better to remove this section and concentrate the paper entirely on ARK1, however I do feel it important to include all data presented, as it will add to an ongoing debate within the community as to the role of LysM-proteins in mediating signaling events in AM symbiosis, with a unique twist – signaling at the PAM.

We concur with the reviewer that the data on LYK1 are less conclusive than those on ARK1 but still would be of interest to the field. We have therefore maintained the description of the data but have moved them to the Supplementary Information.

2. While it is clear that the authors were careful to identify relevant markers to track enrichment of proteins from underrepresented membrane tissues, for example, PAM-specific proteins such as STR and PT6, what is less clear is how carefully non-membrane associated (ie, cytoplasmic) proteins were tracked as a means of estimating the inevitable 'contamination' of these within the membrane-enriched fractions. For example, in looking through the plant (and fungal) proteins in Supp Table 1, I can find some enzymes, trxn factors, uncharacterized proteins that lack canonical signal peptides and TMD, etc, that are unlikely to be membrane-bound (or directly associated with membrane proteins) and are most likely high abundance cytoplasmic proteins. It also seems as though the Blast2Go was used to assess PM enrichment against other membrane proteins, but not contaminating non-membrane proteins. While this in no way invalidates the discovery of ARK, etc, it would be informative to discuss this briefly, and useful to include in Supp Table 1 an in silico assessment of protein localization for all of the proteins listed (eg, wolf psort and TMHMM2), to help discriminate between the genuine membrane proteins and contaminating non-membrane proteins. Along this vein, in pg 8, when the authors identify for example 3640 'membrane' proteins, has this classification been supported by in silico analyses (eg, presence of a signal peptide and TMD) or would it be more correct to classify these as 'proteins extracted from membrane-enriched fractions'? My comments apply equally to the rice root proteins (Suppl Table 7).

We fully agree with the reviewer that contamination is inevitable but would like to stress that we intended to enrich and not to purify membrane proteins, which could be challenging in the absence of information on the physico-chemical properties of the PAM. We feel that including an overview over protein localisation would add little information at this point.

3. While perhaps not surprising, the detection of hundreds of fungal proteins from both colonized maize and rice roots is an intriguing additional data set, but is not discussed at all by the authors. While the focus of the project was to identify PAM-localized plant proteins, presumably the methods employed would have also enriched for components of the fungal membrane and its proteins?? I appreciate space limitations, but it would be worth adding a comment or two as to the nature of these fungal proteins. Do the authors expect the fungal membranes to co-purify with the PAM? and do they believe these fungal proteins to originate from the fungal membrane, or perhaps these are mostly contaminating cytoplasmic proteins? Any evidence of so-called effector proteins that might be associated with the PAM?

*The reviewer raises a point that we have overlooked. We have included an extra worksheet specifically mentioning the fungal proteins in **Supplementary Tables 3 and 7**.*

4. I am concerned about the surprisingly small number of AM-specific proteins that overlap between maize and rice (Fig 2c). Yes, some of large number of non-overlapping proteins will be attributable to species specificity (line 209; pg 10) but I do suspect much of this is due to noise...however this may be a necessary trade-off to achieve the sensitivity required for this approach to be successful.

We understand the reviewer's concern, and wish to respond twofold. (1) The maize and the rice root systems are different, e.g. rice crown roots constitutively have aerenchyma and fine lateral roots are immune against AM fungal colonisation, whereas in maize aerenchyma is missing and all roots are colonised. (2) The rice genome annotation is significantly better than that of maize, the latter thereby sometimes compromising the identification of the correct protein match between the two species in addition to the presence of confounding multiple isoforms. Please also see our reply to reviewer 1, comment 7.

Specific comments

5. Pg 6 – line 111 – yellow pigment is a bit vague. Is anything further known about this pigment? presumably mock inoculated roots do not accumulate this pigment, which raises the question of how the MI control roots were selected for harvest? Were these roots randomly chosen?

The 'yellow pigment' corresponds to an apocarotenoid that specifically accumulates in arbusculated cells. We have amended to text to provide more specific information (page 6, line 111). Regarding the MI root we confirm their random sampling.

6. Pg 7 – line 139 – 'within the distal' I think the authors mean 'within the distal membrane'?

The text has been corrected as suggested (page 6, line 112).

7. Pg 12 – Mol. Characterization of LYK1 - Gomez et al., 2009 presented evidence (also obtained by laser capture microdissection) of a LysM RLK (MtLYR1) that is specifically expressed in arbusculated cells – I am curious whether this is a homologue of LYK1? If so, it would be appropriate to cite this work.

We confirm that LYK1 is not a homologue of MtLYR1. Following classification by the Arrighi et al. (2006), MtLYR1 falls into a separate Clade (II), which also contains NFR/NFP while LYK1 falls into Clade I that lacks NFP/NFR5 and contains LYK3 & NFR1. The closest homologue to OsLYK1 in M. truncatula is LYK10 (Medtr5g033490).

8. Pg 13- lines 301-303, and Suppl Fig 3 – the authors demonstrate that two lines of LYK1 mutants have WT colonization levels at 6 wpi, however it seems odd that they make no comment or present data about colonization at an earlier time point (ie, 3 wpi) and/or no comment about qualitative phenotypes such as formation of arbuscules, etc. The implication in the text of the results and discussion is that the LYK1 mutants exhibited no abnormal phenotype and this may be due to redundancy with other RLKs, however it would be appropriate to clarify whether, for example, an earlier 3 wpi time point was assessed and determined to be comparable with WT.

We indeed assessed both arbuscule morphology and temporal colonisation of lyk1 insertional mutants and did not observe any phenotypic difference relative to the wild type.

We indeed looked at early time points in the null allele lyk1-1 but found no evidence for lower colonisation or morphological alterations of fungal structures. Please see colonization data below.

R. irregularis colonization in the *lyk1-1* mutant allele at 3, 4 and 5 weeks post inoculation (wpi). T, Total colonization; EH, extracellular hyphae; H, hyphopodia; IH, intra-radical hyphae; A, arbuscules; S, spores; V, vesicles, open circles correspond to biological replicates. For statistical analysis Kruskal-Wallis tests with the Holm adjustment method were performed using the agricolae package and selecting the option for "treatment groups formation" with P value set at ≤ 0.05 . The letters above each bar indicate colonization values that were not significantly different in the post hoc pairwise comparisons.

9. Pg 16 – lines 354-356 – the authors complemented the *ark1-1* mutant with the mRFP construct and concluded 'a restoration of wild-type amounts of fungal structures...demonstrating that the ARK1-mRFP fusion protein was functional' however looking at the data, there is statistically significant difference between the colonization profiles of WT plants and *ark1-1* lines complemented with ARK1-mRFP, suggesting only partial complementation. If this is the case, the authors should modify their statement in lines 345-346 accordingly.

*We apologise for this oversight. The text has been amended accordingly to "significantly higher levels of colonization was observed in *ark1-1*^{ARK1-mRFP} compared to *ark1-1*^{EV} control lines (Fig. 4e), demonstrating at least partial functionality of the ARK1-mRFP fusion protein". (Page 15, Line 323-325).*

10. Pg 19 – line 449 – 100mM and 25mM KH₂PO₄?? Surely it would be micromolar?

The concentration has been corrected for μ M (Page 18, Line 414).

11. Fig 1. – last line of caption, correct 'arbuscule'

The figure (new Supplementary Figure 1) legend corrected as suggested.

12. Fig 4 – I'm wondering whether this figure would be better placed in Supplemental?

We agree with the reviewer and have changed it to new Supplementary Figure 5.

13. Fig. 4C – part of the figure is cut off (activation loop)

The figure (now Supplementary Fig. 6c) has been corrected.

Reviewers' Comments:

Reviewer #1:

Remarks to the Author:

The authors have addressed all my previous concerns. In particular, the documentation of the proteomics data has been substantially improved. The authors also added additional microscopic documentation to show that the expression of ARK1 is arbuscule-specific. The TEM localisation provides additional evidence for PAM specificity. While I agree with some of the other reviewers' suggestion for a more detailed functional/mechanistic characterisation of ARK1, I feel that the novelty of the identification of ARK1 as a protein involved in arbuscule maintenance warrants publication; mechanistic characterisation is likely to involve substantial future work. In particular, it was very interesting to see that presence of WT plants in a nurse system rescued the mycorrhizal defects in the ark mutants. In addition, the proteomics data form a significant platform on which the identification of ARK1 was based, and will provide future opportunities for further protein characterisation of other candidate proteins in AM symbioses.

I do have some minor points that should be corrected:

1. In the legend to Table 1, please define APM and MPM.
2. In figure 3c and d, I can see the scale bars on the screen, but they are hard to see in a printed version.
3. Figure 4c: scale bar is missing.
4. Figure 4 legend: please indicate the time points at which the data were collected (I assume 6 wpi, but please specify).
5. Figure 4e: please indicate in the legend that this figure is showing medians (I assume).
6. Figure 4f-h: Again, in the printed version, the arrows are hard to see.
7. For the microscopy images, it would be beneficial to state in the legends how many samples were analysed in each case and how many samples showed similar results (i.e. Fig 2, Fig. 3cd, Fig. 4a-d, 4f-h).
8. The reference list needs some corrections, especially please check that species names are italicised, and that capitalisation and journal names are consistent.

Reviewer #3:

None

We would like to thank reviewer #1 for her/his final comments to our manuscript.

Below, please find our point-for-point reply to final concerns raised by reviewer #1.

REVIEWERS' COMMENTS:

Reviewer #1 (Remarks to the Author):

The authors have addressed all my previous concerns. In particular, the documentation of the proteomics data has been substantially improved. The authors also added additional microscopic documentation to show that the expression of ARK1 is arbuscule-specific. The TEM localisation provides additional evidence for PAM specificity. While I agree with some of the other reviewers' suggestion for a more detailed functional/mechanistic characterisation of ARK1, I feel that the novelty of the identification of ARK1 as a protein involved in arbuscule maintenance warrants publication; mechanistic characterisation is likely to involve substantial future work. In particular, it was very interesting to see that presence of WT plants in a nurse system rescued the mycorrhizal defects in the ark mutants. In addition, the proteomics data form a significant platform on which the identification of ARK1 was based, and will provide future opportunities for further protein characterisation of other candidate proteins in AM symbioses.

I do have some minor points that should be corrected:

1. In the legend to Table 1, please define APM and MPM.

We have updated Table 1 to define APM and MPM as follows: "MPM: Mock Plasma Membrane Proteome; APM: Arbuscular Plasma Membrane Proteome"

2. In figure 3c and d, I can see the scale bars on the screen, but they are hard to see in a printed version.

The thickness of the scale bars in figure 3c and d have been increased to ensure they are visible in a printed version.

3. Figure 4c: scale bar is missing.

The scale bar in Figure 4c has been added.

4. Figure 4 legend: please indicate the time points at which the data were collected (I assume 6 wpi, but please specify).

The time points at which the data were collected have been included in the legend of Figure 4 and all figures where appropriate.

5. Figure 4e: please indicate in the legend that this figure is showing medians (I assume).

We have updated Figure 4e accordingly to indicate that the figure is indeed showing medians.

6. Figure 4f-h: Again, in the printed version, the arrows are hard to see.

To ensure the arrows in Figure 4f-h are visible in print we have increased their thickness.

7. For the microscopy images, it would be beneficial to state in the legends how many samples were analysed in each case and how many samples showed similar results (i.e. Fig 2, Fig. 3cd, Fig. 4a-d, 4f-h).

We have amended all figures legends to indicate the number of roots analyzed and from how many independent biological replicates for representative images. For example the figure legend for Fig 2 now includes: "Representative images of three

roots from five replicate plants are provided."

8. The reference list needs some corrections, especially please check that species names are italicised, and that capitalisation and journal names are consistent. We have amended the reference list as suggested.